# Monkeypox virus genomic accordion strategies

The 2023 monkeypox (mpox) epidemic was caused by a subclade IIb descendant of a monkeypox virus (MPXV) lineage traced back to Nigeria in 1971. Person-to-person transmission appears higher than for clade I or subclade IIa MPXV, possibly caused by genomic changes in subclade IIb MPXV. Key genomic changes could occur in the genome's low-complexity regions (LCRs), which are challenging to sequence and are often dismissed as uninformative. Here, using a combination of highly sensitive techniques, we determine a high-quality MPXV genome sequence of a representative of the current epidemic with LCRs resolved at unprecedented accuracy. This reveals significant variation in short tandem repeats within LCRs. We demonstrate that LCR entropy in the MPXV genome is significantly higher than that of single-nucleotide polymorphisms (SNPs) and that LCRs are not randomly distributed. In silico analyses indicate that expression, translation, stability, or function of MPXV orthologous poxvirus genes (OPGs), including *OPG153*, *OPG204*, and *OPG208*, could be affected in a manner consistent with the established "genomic accordion" evolutionary strategies of orthopoxviruses. We posit that genomic studies focusing on phenotypic MPXV differences should consider LCR variability.

Monkeypox virus (MPXV) is a double-stranded DNA virus classified in genus *Orthopoxvirus* (varidnavirian *Nucleocytoviricota*: *Poxviridae*: *Chordopoxvirinae*) along with other viruses, such as vaccinia virus (VACV) and variola virus (VARV) that also can infect humans[1]. MPXV causes "monkeypox (mpox)" (World Health Organization International [WHO] Classification of Diseases, Eleventh Revision [ICD-11] code 1E71)[2].

First encountered in 1958 in crab-eating macaques imported to Belgium[3], MPXV has caused sporadic human disease outbreaks since the 1970s in Eastern, Middle, and Western Africa, totaling approximately 25,000 cases (case fatality rate 1–10%)[4], and also sporadic disease outbreaks among wild monkeys and apes[5,6]. Exposure to MPXV animal reservoirs, in particular rope squirrels and sun squirrels, is a significant risk factor of human infections[7].

Since May 2022, multiple European countries have reported a continuously increasing number of MPXV infections and associated disease, including clusters of cases associated with potential super-spreading events in Belgium, Spain, and the United Kingdom (UK). As of January 10, 2024, a total of 94,274 cases had been reported in 118 countries/territories/areas in all six WHO regions. While the number of new cases has decreased over time, cases of the disease are still occurring among vaccinated individuals. Therefore, as the duration of the virus's circulation in humans increases, the risk of emergence of a more transmissible variant capable of causing larger outbreaks escalates.

Phylogenetically, historic MPXV isolates cluster into two clades[8], designated I and II[9,10]. Clade I viruses are considered more virulent and transmissible than clade II viruses[8,9,11]. The viruses of the 2022 epidemic belong to subclade IIb[12–14], a line of descent of MPXV that had been circulating in Nigeria, likely since 1971[15].

The clinical presentation of mpox caused by MPXV clade I or subclade IIa includes fever, headache, lymphadenopathy, and/or malaise, followed by a characteristic rash that progresses centrifugally from maculopapules via vesicles and pustules to crusts that may occur on the face, body, mucous membranes, palms of the hands, and soles of the feet[16]. The clinical presentation of subclade IIb infection diverges

✉e-mail: gustavo.palacios@mssm.edu

from classical mpox by having a good prognosis, self-limiting but infectious skin lesions (typically emerging at and restricted to the genital, perineal/perianal, and/or peri-oral areas) before the development of fever, lymphadenopathy, and malaise. Generalized disease usually manifests with a rash that has not been widely observed in the current outbreak. Human-to-human transmission is substantially higher in outbreaks associated with subclade IIb MPXV than those caused by clade I and subclade IIa[17–21]. The $R_0$ for MXPV IIb among men who have sex with men (MSM) is higher than 1. Transmission may be catalyzed by a decrease in protection associated with the VARV/smallpox vaccination campaign that ended in 1980[22,23]. Furthermore, a change in transmission route may be the cause of the difference in clinical presentation and pathogenesis as was shown in animal models[24].

Orthopoxvirus infections are classified as systemic or localized[25]. The involved orthopoxvirus and the immune status of the host are determinants of generalized or localized infection. Different mechanisms of virion entry and egress, as well as virus-encoded host restriction factors, also play pivotal roles in determining the clinical manifestation of infection[26–30]. Localized usually means that signs are restricted to the site of viral entry, which is the most common clinical presentation described in the 2022 mpox outbreak. Changes in the genome of the current MPXV variant, such as gene loss[31] may explain both trends.

The MPXV genome is a linear, ≈197-kb-long double-stranded DNA with covalently closed hairpin ends. The genome's densely packed orthologous poxvirus genes (OPGs)[32] are distributed over a central conserved region ("core") and flanking terminal regions, each of which ends in identical but oppositely oriented ≈6.4-kb-long terminal inverted repetitions (ITRs). Roughly 193 open reading frames (ORFs) encode proteins with ≥60 amino-acid residues. "Housekeeping" proteins involved in MPXV transcription, replication, and virion assembly are encoded by OPGs located in the central conserved region, whereas proteins involved in host range and pathogenesis are mostly encoded by OPGs located in the terminal regions[33]. Like all orthopoxvirus genomes, the MPXV genome contains numerous tandem repeats in the ITRs as well as nucleotide homopolymers all over the genome[33–36]. However, other similar structures through the MPXV genome were observed in the form of short tandem repeats (STRs). Moreover, initial observations appear to indicate that these STRs (which may consist of dinucleotide, trinucleotide, or more complex palindromic repeats) are localized in areas where more variation is observed, suggesting a crucial role in MXPV biology and evolution.

Orthopoxviruses rapidly acquire higher fitness by massive gene amplification (genome expansion) when encountering severe bottlenecks in vitro. This amplification, akin to gene reduplication in organismal evolution, enables gene copies to accumulate mutations, potentially resulting in protein variants that can overcome the bottlenecks. Subsequent gene copy reduction (genome contraction) offsets the costs associated with increasing genome length, thereby retaining the adaptive mutations[37]. Orthopoxviruses also rapidly adapt to selective pressures by single-nucleotide insertions (genome expansion) or deletions (genome contractions) within poly-A or poly-T stretches, resulting in easily reversible gene-inactivating or re-activating frameshifts[38]. These rhythmic genome expansions and contractions are referred to as "genomic accordions" at the gene and base level[37]. Given the overall conservation of STRs in orthopoxvirus genomes, we hypothesized that their variation could be a third type of genomic accordion and that, overall, this type of adaptation (which we designate here as low-complexity regions [LCRs]), rather than single-nucleotide polymorphisms (SNPs), could be the key to understanding the unusual epidemiology of 2022 subclade IIb MPXV. We do not know whether this different epidemiology is due intrinsically to the virus, to different host behavior, and/or different transmission routes, but all should influence the composition of the MPXV genome, as different selective and purifying pressures would necessarily leave marks on the affected viral effectors.

In this study, we provide a comprehensive genomic characterization of LCRs during the mpox outbreak. Our analysis establishes the fact that LCRs exhibit a non-random distribution across the genome. Moreover, we demonstrate that LCRs display higher entropy compared to SNPs. Importantly, our findings highlight three specific gene candidates that warrant further investigation in relation to transmissibility and/or adaptation. As a result, we propose a focused examination of LCR variability in future MPXV genomic analyses. The distinctive characteristics observed in LCRs emphasize their potential significance in understanding the dynamics of the currently circulating viral clades and suggest promising avenues for targeted research.

## Results

### De novo assembly of subclade IIb lineage B.1 MPXV genome sequence 353R

Using a template-based mapping approach, shotgun metagenomic short-read-based sequencing of nucleic acids in vesicular lesion swabs from Spanish mpox patients resulted in the determination of 48 MPXV consensus genome sequences with at least 10X read depth. A median of 39,697,742 high-quality reads per swab (maximum = 111,030,976; minimum = 7,780,032) were obtained using a NovaSeq 6000 Illumina sequencer. Although 98.12% of the reads were assigned as being of human origin, a median of 74,085 MPXV reads (maximum = 27,516,891; minimum = 30,854) sufficed to cover >99% of the genome (Supplementary Data 1). Oxford Nanopore MinION Mk1B sequencing of swab 353R generated 410,050 reads, with a median read length of 420 nucleotides and a median read quality of 10.9 Phred.

Read mapping indicated that LCRs of the MPXV genome were mostly unresolved. More importantly, those results were biased by the reference genome used as a scaffold. In general, the observation is that LCRs are resolved by reference mapping software tools "following" the pattern provided in the scaffold reference genome instead of reporting the actual pattern (Figure S1a). To determine the actual LCR sequence, we explored assembly strategies generally used for resolving eukaryotic genomes, which mostly combine different sequencing technologies. To increase the chances of success, we applied these technologies to an mpox patient sample with a high proportion of high-quality viral reads (swab 353R). The de novo assembly obtained from Illumina NovaSeq (2×150-bp pair-ended reads), MiSeq (2×300-bp pair-ended reads), and Oxford Nanopore MinION Mk1b sequencing generated 3, 2, and 1 contigs belonging to MPXV, covering 97, 97, and 101% of the MPXV-M5312_HM12_Rivers sequence, respectively (Fig. 1). The Oxford Nanopore MinION Mk1b de novo assembly had a median depth of coverage of 196X.

### Characterization and validation of non-randomly distributed LCRs in MPXV genome sequence 353R

We applied a systematic approach for LCR discovery to the MPXV 353R sequence that resulted in the identification of 21 LCRs (13 STRs, 8 homopolymers; Table 1 and Supplementary Data 2). Two pairs of LCRs (1/4 and 10/11) are located in the ITRs and are identical copies in reverse-complementary form.

In general, LCRs were resolved using the assembly obtained from single-molecule sequencing and further validated using short-read sequencing since most sequences were 13–67 bp long and therefore were covered by reads from each side or flanking region without mismatches (Figure S1b; Supplementary Note 1). All LCRs were validated this way with the exception of LCR pair 1/4 (256 bp) and LCR3 (468 bp), which were only resolved with single-molecule sequencing reads due to their lengths (Table 2).

LCR3 contains a complex tandem repeat with the sequence ATAT[ACATTATAT]n with $n = 52$ in the MPXV 353R genome sequence. No publicly available MPXV genome sequence contains a tandem repeat

of similar length (e.g., likely due to the limitations of the high-throughput sequencing techniques used for their characterization). However, applying the analysis to 35 publicly available MPXV National Center for Biotechnology Information (NCBI) Sequence Read Archive (SRA) datasets of single-molecule raw reads allowed us to resolve the LCR3 of some (Fig. 2a). Fifteen datasets revealed supporting long reads that include both LCR3 flanking regions. Interestingly, four subclade IIb lineage B.1 MPXV sequences associated with the 2022 mpox epidemic and available in SRA have 54–62 repeats in LCR3. Their number of repeats ($n$) separate these sequences from 2018–2019 subclade IIb

lineage A sequences that have 12–42 repeats in LCR3, indicating LCR3 as a region of genomic instability and high variability.

LCR pair 1/4 contains a tandem repeat with the sequence [AACTAACTTATGACTT]$_n$ with $n = 16$ in the MPXV 353R genome sequence. Instead, the sequences of the subclade IIb lineage B.1 MPXV isolate MPXV_USA_2022_MA001 and lineage A reference isolate MPXV-M5312_HM12_Rivers LCR pair 1/4 have $n = 8$ (Table 3). Inclusion of NCBI SRA datasets into the analysis confirmed the $n = 16$ value (Fig. 2b). In addition, the analysis of the SRA datasets revealed subclade IIb lineage-specific repeat differences in LCR pair 1/4. Lineage A.1 virus genomes

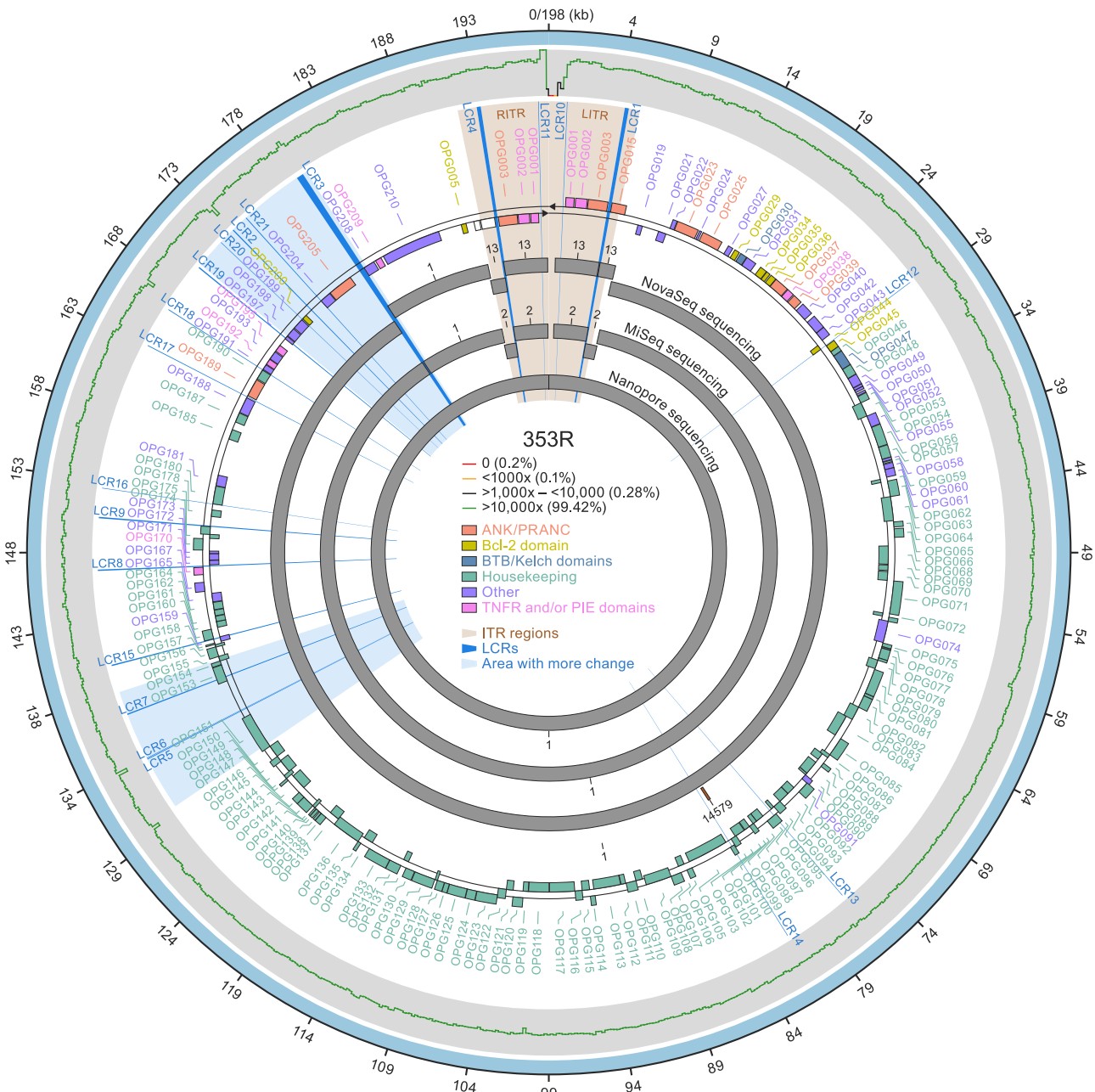

**Fig. 1 | De novo assembly of subclade IIb lineage B.1 monkeypox virus (MPXV) genome sequence 353R.** A visual representation of the fully annotated MPXV isolate 353R genome (based on the subclade IIb lineage A reference isolate MPXV-M5312_HM12_Rivers genome sequence annotation). Shown are (from the outside to the inside): high-quality genome (HQG) hybrid assembly (wide outer light blue ring); sequencing coverage distribution graph (thin ragged line [green: ≥10,000x, 99.42%; black: 1000x–10,000, 0.28%; orange: <1000x–10, 0.1%; red: <10–0, 0.2%]); orthologous poxvirus gene (OPG) annotations according to the standardized

nomenclature[32] (lettering and shaded boxes [orange: ANK/PRANC (N-terminal ankyrin protein with PRANC domain) inverted terminal repetition [ITR] regions; gold: Bcl-2 domain; blue: BTB/Kelch domains; green: housekeeping; purple: other; pink: TNFR and/or PIE domains]; contigs from NovaSeq, MiSeq, and nanopore sequencing (wide inner gray rings). Additionally, radial lines and shading that originate in the center and reach outward on the white background indicate low-complexity regions (LCRs; royal blue) and areas with more change (light blue).

**Table 1 | Low-complexity regions (LCRs) in monkeypox virus (MPXV) genome sequence 353R**

| Name | Location start[a] | Location end[b] | Repeat unit[c] | Pattern[d] | Nearest OPG[e] | Type of LCR[f] | Relative position to the OPG[g] | Distance in bp[h] | Copenhagen notation[i] | Vaccinia virus (VACV) notation[j] | Comments |
|---|---|---|---|---|---|---|---|---|---|---|---|
| LCR1 | 5369 | 5624 | 16 | [AACTAACTTATGACTT]n | OPG003 (ITR) | STR | Downstream | 72 | Cop-C19L | NA | |
| LCR1 | 5369 | 5624 | 16 | [AACTAACTTATGACTT]n | OPG015 (ITR) | STR | Upstream | 35 | CPXV-O17 | NA | |
| LCR2 | 174,063 | 174,112 | 2 | [ATAT]n | NA | STR | Downstream | 46 | Cop-B16R | B14R | |
| LCR3 | 179,872 | 180,345 | 9 | ATAT [ACATTATAT] | OPG208 | STR | ATG Start/Promoter | 21 | Cop-K2L | B19R | SPI-1 apoptosis inhibition |
| LCR4 | 193,504 | 193,759 | 16 | [AAGTCATAAGTTAGTT]n | OPG003 (ITR) | STR | Downstream | 72 | Cop-C19L | NA | |
| LCR4 | 193,504 | 193,759 | 16 | [AAGTCATAAGTTAGTT]n | OPG015 (LITR) | STR | Upstream | 35 | CPXV-O17 | NA | |
| LCR5 | 133,895 | 133,918 | 1 | [T]n | MPXVgp137 | homopolymer | Upstream | 889 | Cop-A25L | A27L | Fragmented gene area |
| LCR6 | 133,980 | 133,989 | 10 | [CAATCTTTCT]n | MPXVgp137 | STR | Upstream | 818 | Cop-A25L | A27L | |
| LCR7 | 137,319 | 137,375 | 3 | [ATC]n | OPG153 | STR | Inside ORF | NA | Cop-A28L | A26L | Attachment MVs/laminin |
| LCR8 | 147,655 | 147,718 | 5 + 7 | [ATATTT]n, [ATTTT]n, [ATATTTT]n [ATTTT]n, [ATATTTT]n [ATTTT]n [ATATTT]n | OPG171 | STR | Upstream | 75 | Cop-A42R | A42R | |
| | | | | | OPG170 | STR | Upstream | 70 | Cop-A41L | A41L | |
| LCR9 | 151,350 | 151,417 | 9 | [TATGAAG]n, [GATATGAT]n [GATATGATG]n, [GATATGAT]n | OPG176 | STR | Upstream | 12 | Cop-A46R | A47R | |
| LCR10 | 197,830 | 197,842 | 1 | [T]n | OPG001 (ITR) | homopolymer | Downstream | 225 | NA | NA | |
| LCR11 | 1286 | 1298 | 1 | [T]n | OPG001 (ITR) | homopolymer | Downstream | 225 | NA | NA | |
| LCR12 | 29,326 | 29,364 | 1 | [A]n | OPG044 | homopolymer | Inside ORF | NA | Cop-K7R | B15R | C-terminal position |
| LCR13 | 76,896 | 76,904 | 1 | [T]n | OPG097 | homopolymer | Inside ORF | NA | Cop-L3L/L4R | L3L/L4R | |
| LCR14 | 81,658 | 81,666 | 1 | [T]n | OPG104 | homopolymer | Inside ORF | NA | Cop-J5L | L5L | Essential for viral replication |
| LCR15 | 140,911 | 140,977 | 9 | [ATAACAATT]n, [ATAATAATT]n [ATAATTGTT]n, [ATAATTGTT]n | OPG159 | STR | Inside ORF | NA | Cop-A3L | A33L | PKR inhibitor candidate? / C-terminal position |
| LCR16 | 153,457 | 153,465 | 1 | [A]n | OPG180 | homopolymer | Upstream | 15 | Cop-A50R | A50R | |
| LCR17 | 163,979 | 164,003 | 4 | [TAAC]n | OPG188 | STR | Downstream | 82 | Cop-B2R | B4R | |
| LCR18 | 166,865 | 166,920 | 7 | [AATAATT]n | OPG190 | STR | Downstream | 15 | Cop-B5R | B6R | |
| LCR19 | 170,508 | 170,563 | 6 | [GATACA]n | OPG197 | STR | Inside ORF | NA | Cop-B11R | B11R | Hypothetical protein |
| LCR20 | 172,868 | 172,876 | 1 | [T]n | OPG199 | homopolymer | Downstream | 56 | Cop-K2L | SPI-2/B12R | |
| LCR21 | 175,299 | 175,357 | 6 | [GATGAA]n | OPG204 | STR | ATG Start/Promoter | NA | Cop-B19R | B16R | Alternative ATG repeat start |

Short tandem repeats (STRs) are described using nucleotide base-pair coordinates with reference to the high-quality genome (HQG) sequence (ENA Accession #OX044336). Listed are the number of repeats for this particular genome, identification of the nearest annotated orthologous poxvirus gene (OPG), type of LCR (STR or homopolymer), position of the LCR to the nearest gene, and distance of the LCR to the nearest gene. OPG notations follow the standardized nomenclature[32], vaccinia virus (VACV) Copenhagen strain and classical VACV gene notations are shown in addition to enable comparisons. NA not applicable.
[a]nucleotide base coordinate in reference HQG (Genbank #OX044336).
[b]nucleotide base coordinate in reference HQG (Genbank #OX044336).
[c]number of repeat units in the HQG (Genbank #OX044336).
[d]description of the pattern of the LCR in representative MPXV, in which n is the number of repeats for this particular genome.
[e]identification according to Senkevich et al. of nearest identified gene; new notation.
[f]type of LCR: short tandem repeats or homopolymer.
[g]position of the LCR to the nearest gene.
[h]distance of the LCR to the nearest gene.
[i]notation of the gene in the VACV Copenhagen strain.
[j]notation of the gene in the VACV Western Reserve strain.

**Table 2 | Low-complexity region (LCR) validation and entropy-level intra-host analysis in monkeypox virus (MPXV) genome sequence 353R**

| Name | Repeat Unit | Pattern HQ | Number of repeats HQG | Nearest Gene | Variation | Type of Variation | Entropy threshold > 0.03 | Resolved correctly in RefSeq | Nanopore | MiSeq | NovaSeq | # Supporting Reads MiSeq | # Supporting Reads NovaSeq |
|---|---|---|---|---|---|---|---|---|---|---|---|---|---|
| LCR4 | 16 | TAGTCATAAGTAGTT [AAGTCATAAGTTAGTT]$_{15}$ | 16 | OPG003 (ITR) | NR | Length | NA | No$^b$ | Yes | No | No | NA | NA |
| LCR3 | 9 | ATAT [ACATTATAT]$_{52}$ | 52 | OPG208 | Yes | Length | NA | Yes | Yes | No | No | NA | NA |
| LCR1 | 16 | [AACTAACTTATGACTT]$_{15}$ AACTAACTTATGACTA | 16 | OPG003 (ITR) | NR | Length | NA | No$^b$ | Yes | No | No | NA | NA |
| LCR2 | 2 | [AT]$_{25}$ | 25 | NA | Yes | Length | 1.66 | No | Yes | Yes | Yes | 768 | 90 |
| LCR5 | 1 | [T]$_{24}$ | 24 | OPG152 | Yes | Length | 1.535 | Yes | No | No | Yes | NA | 112 |
| LCR10 | 1 | [T]$_{13}$ | 13 | OPG001 (ITR) | Yes | Length | 0.63 | No$^b$ | Yes | No | No | 6561 | 11,945 |
| LCR11 | 1 | [T]$_{13}$ | 13 | OPG001 (ITR) | Yes | Length | 0.627 | No$^b$ | Yes | Yes | Yes | 6448 | 11,589 |
| LCR21 | 6 | [GATGAA]$_4$ GATGA | 4.5 | OPG204 | Yes | Mutation | 0.207 | Yes | Yes | Yes | Yes | 6578 | 6661 |
| LCR7 | 3 | [ATC]$_{14}$ TATGAT [ATC]$_3$ | 19 | OPG153 | Yes | Length | 0.181 | Yes | Yes | Yes | Yes | 4541 | 6607 |
| LCR9 | 9 | [TATGAAG]$_1$ [GATATGAT]$_1$ [GATATGATG]$_{15}$ [GATATGAT]$_1$ | 8 | OPG176 | No | NA | 0 | Yes | Yes | Yes | Yes | 5208 | 5737 |
| LCR8 | 5 + 7 | [ATATTT]$_1$ [ATTTT]$_1$ [ATATTTT]$_3$ [ATTTT]$_1$ [ATATTTT]$_2$ [ATTTT]$_1$ [ATATTTT]$_1$ | 10 | OPG171 | No | NA | 0 | Yes | Yes | Yes | Yes | 6581 | 6790 |
| LCR6 | 10 | [CAATCTTTCT]$_1$ | 1 | OPG152 | Yes$^a$ | NA | 0 | No$^a$ | Yes | Yes | Yes | 4884 | 12,930 |
| LCR20 | 1 | [T]$_9$ | 9 | OPG199 | No | NA | 0 | Yes | Yes | Yes | Yes | 10,106 | 13,315 |
| LCR19 | 6 | GATTCA [GATACA]$_8$ GAT | 9.3 | OPG197 | No | NA | 0 | Yes | Yes | Yes | yes | 4119 | 4685 |
| LCR18 | 7 | [AATAATT]$_3$ AATAA | 3 | OPG190 | No | NA | 0 | Yes | Yes | Yes | Yes | 9755 | 11,838 |
| LCR17 | 4 | [TAAC]$_{16}$ T | 6.1 | OPG188 | No | NA | 0 | Yes | Yes | Yes | Yes | 7388 | 9474 |
| LCR16 | 1 | [A]$_9$ | 9 | OPG180 | No | NA | 0 | Yes | Yes | Yes | Yes | 10,340 | 16,044 |
| LCR15 | 9 | [ATAACAATT]$_4$ [ATAATTGTT]$_1$ [ATAATAATT]$_1$ [ATAATTGTT]$_1$ | 7 | OPG159 | No | NA | 0 | Yes | Yes | Yes | Yes | 7067 | 6569 |
| LCR14 | 1 | [T]$_9$ | 9 | OPG104 | No | NA | 0 | Yes | Yes | Yes | Yes | 7819 | 12,521 |
| LCR13 | 1 | [T]$_9$ | 9 | OPG097/ 098 | No | NA | 0 | Yes | Yes | Yes | Yes | 7480 | 12,126 |
| LCR12 | 1 | [A]$_9$ | 9 | OPG044 | No | NA | 0 | Yes | Yes | Yes | Yes | 9789 | 13,592 |

Listed are the type and number of supporting reads for each LCR. Definitions of quality: Yes, LCR is found entirely in the assembly in one contig; no, LCR is not assembled with the reported method. All LCRs with entropy levels above 0.15 are shaded in gray. OPG orthologous poxvirus gene.

$^a$LCR6 is a 10-bp repeat that was reported early in the outbreak as an insertion (https://virological.org/t/first-german-genome-sequence-of-monkeypox-virus-associated-to-multi-country-outbreak-in-may-2022/812). In our dataset, we have not seen any variation in this area.

$^b$LCR pairs 1/4 and 10/11 are located in ITRs. Given that no read covering this area reached a unique are outside of the ITRs, we cannot technically state that we solved the repeat. Nonetheless, the ITRs should be identical based on the know poxvirid replication mode.

**a**

**b**

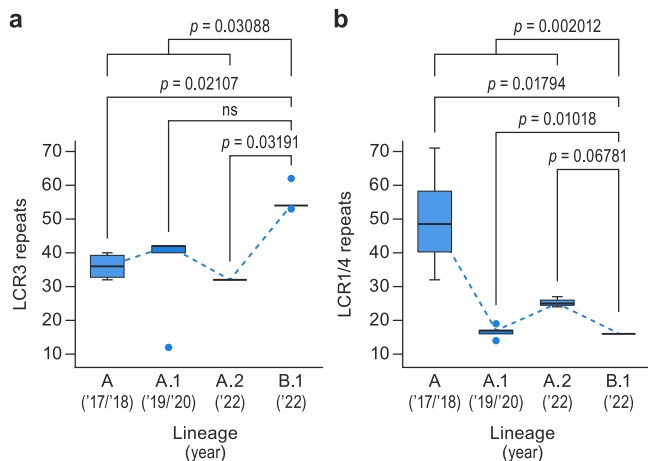

**Fig. 2 | Characterization and validation of non-randomly distributed low-complexity regions (LCRs) in monkeypox virus (MPXV) genome sequence 353R. a** LCR3 sequence validation using MPXV 353R nanopore sequencing data and 15 additional raw data sequencing reads downloaded from the National Center for Biotechnology Information (NCBI) Sequence Read Archive (SRA). Box plots representing median (middle line), 25th and 75th percentile (box), and 5th and 95th percentile (whiskers) outliers are plotted as single dots. Blue dashed lines represent trends of LCR3 repeats in MPXV lineages ($n = 16$ sequences). (Wilcoxon rank-sum test $p$-values are indicated in the graph.) **b** LCR pair 1/4 sequence validation using MPXV 353R nanopore sequencing data and 20 additional raw data sequencing reads downloaded from NCBI SRA. Box plots representing median (middle line), 25th and 75th percentile (box), and 5th and 95th percentile (whiskers) outliers are plotted as single dots. Blue dashed lines represent trends of LCR1/4 repeats in MPXV lineages ($n = 21$ sequences). (Wilcoxon rank-sum test $p$-values are indicated in the graph.) Detailed information on the represented materials, along with originator and epidemiological data, is provided in Supplementary Data 6.

are polymorphic, having 14–19 repeats; lineage A.2 viruses have 23–26 repeats; and lineage A viruses have 32–71 repeats. In contrast, lineage B1 viruses consistently have 16 repeats. While LCR3 appears to have "increased" in length since the spillover, LCR pair 1/4 appears to be decreasing in length, thus behaving like an "accordion" over time.

The subclade IIb lineage B.1 353R and MPXV_USA_2022_MA001 genome sequences have the same 67 SNPs called against the subclade IIb lineage A reference isolate MPXV-M5312_HM12_Rivers genome sequence (Supplementary Data 7). Additionally, the 353R sequence has two additional paired SNPs in the left and right ITRs (5595G→A; 191,615C→T compared with the MPXV-M5312_HM12_Rivers sequence) that result in the introduction of a stop codon in *OPG015*. We observed this variation in only two other patient samples among our dataset. The 353R and MPXV_USA_2022_MA001 sequences also differ by two number of insertion (ins) or deletion (del) of bases (indels) at positions 133,077 and 173,273, respectively, which correspond to differences in LCR2 and LCR5, respectively. As a result of the accurate resolution of the LCRs, the genome of 353R is 1342 bp longer compared with the MPXV_USA_2022_MA001 sequence and 1338 bp compared with the MPXV-M5312_HM12_Rivers sequence. Most of the variation is due to differences in the length of LCR pair 1/4 and LCR3, along with minor length differences in LCR2, LCR5, and LCR pair 10/11 (Table 3). In general, the number of repeats (*n*) found with the hybrid assembly approach we used here doubled the length of LCRs.

Based on the higher resolution of the MPXV 353R genome sequence, in particular regarding LCRs, we propose this sequence as the new MPXV high-quality genome (HQG) reference sequence[39].

## LCRs are not associated with defective genomes

Sequencing MPXV DNA directly from skin lesions has the advantage of avoiding artifacts arising by virus passage in cell culture, but defective

virus genomes that are unable to replicate might be overrepresented. Consequently, identified LCRs or SNPs might not be important in MPXV pathogenesis as they could be part of defective virus genomes rather than the genome of replicative virus. To rule out this scenario, we analyzed our genomic data (Supplementary Data 4) to evaluate whether the variability observed could be the result of proliferation of defective MPXV genomes.

Defective MPXV genomes may arise naturally in various ways (e.g., via mutations resulting in stop codons or nucleotide insertions or deletions that interrupt ORFs). We hypothesized that insertions or deletions of entire codons (nucleotide triples) would be significantly less likely to occur just by chance than insertions or deletions of single nucleotides. Therefore, we re-analyzed all majority and minority alleles to identify these types of changes. No stop codons were detected in any allele in the LCRs inside of genes, enabling us to quickly rule out this scenario. We then focused on variants leading to LCR length variation (e.g., due to insertions or deletions). None of the seven LCRs located inside of genes caused major allele frameshifts. Among minor alleles, only changes in LCR12 (homopolymer, three samples, frequency range of 0.08–0.26) and LCR14 (homopolymer, allele frequency of 0.037) were observed; neither was related to changes in coding regions. Further, we relaxed the inclusion criteria to consider all alleles occurring at a frequency equal or higher than 0.03 regardless of the depth. This enabled us to perform a statistical comparison among these seven LCRs and others located in non-coding regions of the genome (Figure S6). Among the seven LCRs detected, five, including LCR3, LCR7, and LCR21 (identified as primary candidates driving adaptation), evolved solely as changes in codons (Figure S6). These analyses collectively enabled us to confidently rule out polymerase slippage and defective virions as drivers of the observed variability in LCRs.

## LCRs are non-randomly distributed in the monkeypox virus (MPXV) genome

We compared the distribution of LCRs between the different major functional protein OPG groups following the classification[32]. Differences between functional groups were statistically significant (Kruskal–Wallis test, $\chi^2$ $p$-value < 0.001). Pairwise analysis demonstrated that the functional group "Housekeeping genes/Core" (orthopoxvirus genomic central conserved region) includes LCRs at a significantly lower frequency (multiple pairwise-comparison Wilcoxon test) than functional groups "ANK/PRANC" (false discovery rate [FDR]-corrected $p$-value < 0.0001), "Bcl-2 domain" (corrected $p$-value = 0.04), and "Accessory/Other" (FDR-corrected $p$-value < 0.0001) (Fig. 3a). These analyses indicate that LCRs in orthopoxvirus genomes are non-randomly distributed and that there is a significant purifying selection force against introducing LCRs in central conserved region areas.

Next, we compared the degree of diversity among the 21 identified LCRs with the observed SNP variability that had been the focus of the field. In the 353R HQG sequence, LCRs 2, 5, 7, 10, 11, and 21 had intra-host genetic diversity, with entropy values that ranged from 0.18 (LCR7) to 1.66 (LCR2), with an average of 0.81 and a standard deviation (SD) of 0.64 among them (Table 2). Only five nucleotide positions (1285; 6412; 88,807; 133,894; and 145,431) had intra-host genetic diversity at the SNP level. The entropy values ranged from 0.17 (position 133,894) to 0.69 (position 6412), with an average of 0.38 and a SD = 0.21 among them. Interestingly, a student's t-test revealed a significantly higher level of diversity in LCRs than in SNPs ($p$-value = 0.021; Fig. 3b).

Then, we characterized, collected, and compared the allele frequencies for all LCRs from all dataset samples (Supplementary Data 3) applying the filters described above. Our analyses revealed that the average inter-sample Euclidean distances at LCRs ranged from 0.05 (LCR21) to 0.73 (LCR2) (Fig. 3c). We found statistically significant

**Table 3 | Comparison of monkeypox virus (MPXV) genome sequence 353R with reference sequences**

| | MPXV-M5312_HM12_Rivers | MPXV_USA_2022_MA001 | 353R |
|---|---|---|---|
| **Genome length** | 197,209 | 197,205 | 198,547 |
| **SNPs[a]** | NA | 67 | 69 |
| **Indels[a]** | NA | 10del 7ins | 11del 6 ins |
| **Homopolymeric sites[b]** | 408 | 405 | 399 |
| **Unique SNPs** | NA | 0 | 2 |
| **LCR pair 1/4** | 8 | 8 | **16** |
| **LCR2** | 22 | 24 | **25** |
| **LCR3** | 18 | 16 | **52** |
| **LCR5** | 25 | 28 | **24** |
| **LCR6** | **2** | 1 | 1 |
| **LCR7** | **19** | 17.6 | 17.6 |
| **LCR8** | 10 | 10 | 10 |
| **LCR9** | **8** | 6 | 6 |
| **LCR pair 10/11** | 17 | 14 | **13** |
| **LCR12** | 9 | 9 | 9 |
| **LCR13** | 9 | 9 | 9 |
| **LCR14** | 9 | 9 | 9 |
| **LCR15** | 7 | 7 | 7 |
| **LCR16** | 9 | 9 | 9 |
| **LCR17** | 6.1 | 6.1 | 6.1 |
| **LCR18** | 3.5 | 3.5 | 3.5 |
| **LCR19** | 9.3 | 9.3 | 9.3 |
| **LCR20** | 9 | 9 | 9 |
| **LCR21** | 4.5 | 4.5 | 4.5 |

Low-complexity region (LCR) repetitions for each genome are indicated. Discrepant number (*n*) of LCR repeats are given in bold. *indels* number of insertion (ins) or deletion (del) of bases; *LCR* low-complexity region, *SNP* single-nucleotide polymorphism.
[a]SNPs and indels vs MPXV-M5312_HM12_Rivers,
[b]homopolymers >8 nt in length; bold values indicate discrepant number (n) of LCR repeats.

differences between LCRs (Kruskal–Wallis $\chi^2$ *p*-value < 0.001). More specifically, multiple pairwise comparison Wilcoxon test results showed that all LCRs have significantly different levels of inter-sample distances (FDR-corrected *p*-values < 0.001), except in case of the LCR10 versus LCR11 (FDR-corrected *p*-value = 0.48) and LCR2 versus LCR5 (FDR-corrected *p*-value = 0.25) (Fig. 3c). Average distances in SNPs were 0.0018–0.4168. Our randomization tests revealed that all LCRs have a significantly higher level of inter-sample diversity than the SNPs (all FDR-corrected *p*-values < 0.05) (Fig. 3c). These analyses uncovered that most of the variability in the MPXV genome is located in LCRs. Consequently, we propose that studies focusing on MPXV genomic sequence variation should include LCRs as a possible source of meaningful variation.

### LCRs might be more phylogenetically informative than SNPs for inter-host sequence analysis
Analysis of only two mpox patient samples (353R and 349R) resulted in sufficient sequence coverage information to enable allele frequency comparison in most LCRs (Fig. 4a). Their side-by-side comparison revealed differences in allele frequency in some of them (LCR2, LCR5, and LCR pair 10/11) (Fig. 4b). The sequence coverage achieved with the remaining samples only enabled to unequivocally resolve an LCR subset (i.e., covering both flanking regions: LCRs 2, 5, 7, 8, 9, and pair 10/11). LCR8 and LCR9 were identical across the entire sample set. However, LCR7 and LCR pair 10/11 had considerable intra-host variation, as well as differences in the preponderant allele (LCR pair 10/11) between samples (Fig. 4c).

Phylogenetic (Figure S2a) and haplotype network (Figure S2b) analyses of the sequences determined from the mpox patient samples yielded limited information regarding the outbreak. Most sequences were highly similar and are, therefore, part of the basal ancestral MPXV subclade IIb lineage B1 node. Some sequences formed supporting clusters. Sequences clustered into groups as follows (see also Supplementary Data 4).

- group 1 (lineage B.1): sequences from patients 395, 399, 441, and Floridian MPXV isolate MPXV_USA_2022_FL002 (GenBank #ON676704);
- group 2 (lineage B.1): sequences from patients 347, 352R, 353R, 416, and Spanish MPXV isolate MPXV/ES0001/HUGTiP/2022 (GenBank #ON622718); all share a stop-codon mutation in *OPG015*;
- group 3 (lineage B.1.3): sequences from patient 2,369 formed with Slovenian MPXV isolate SLO (GenBank #ON609725.2), French isolate MPXV_FR_HCL0001_2022 (GenBank #ON622722), and 38 other sequences worldwide; defined by NBT03_gp174 mutation G190,660A, resulting in an R84K amino-acid residue change;
- group 4 (lineage B.1): sequences from patients 417 and 2437;
- group 5 (lineage B.1.1): sequences from patients 698, 1300, 2388, 2428, German MPXV isolate MPXV/Germany/2022/RKI01 (GenBank #ON637938.1), and 97 other sequences worldwide; defined by *OPG094* (*Cop-G9R*) mutation G74,360A, resulting in an R194H amino-acid residue change); and
- group 6 (lineage B.1): sequences from patients 2309 and 2317.

Only one epidemiological link among group samples was identified; patients 395 and 399 were sexual partners who attended events together in both Portugal and Spain.

In summary, there appears to be limited value in full-genome SNP characterization for transmission analysis.

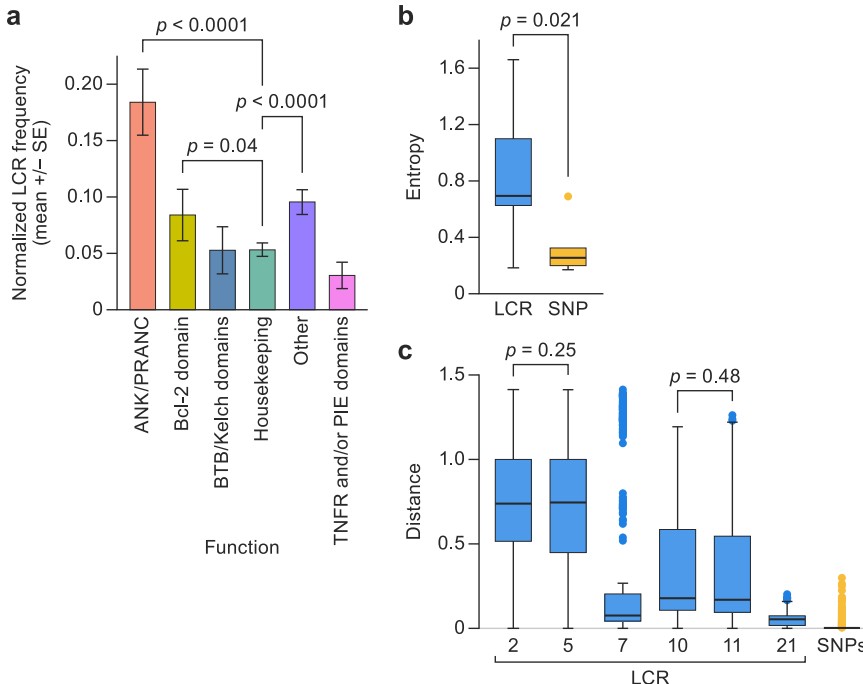

**Fig. 3 | Low-complexity regions (LCRs) are non-randomly distributed in the monkeypox virus (MPXV) genome. a** Frequencies (mean +/- standard error) at which LCRs occur in orthologous poxvirus genes (OPGs) of different functional groups are represented in different colors (*n* = 236 sequences). Shown are functional classes in which pairwise comparisons had a significantly different frequency than other groups. (Multiple pairwise Wilcoxon test false discovery rate [FDR] corrected *p*-value < 0.05 and specific *p*-values are indicated in the graph.) **b** Entropy value distribution for short tandem repeats (STRs) in LCRs (left) and single-nucleotide polymorphisms (SNPs; right). Box plots representing median (middle line), 25th and 75th percentile (box), and 5th and 95th percentile (whiskers) outliers are plotted as single dots; blue = LCR, yellow=SNPs. (Student's t-test-specific *p*-values are indicated in the graph.) **c** Distributions of the pairwise inter-sample Euclidean distances for each STR in LCRs (Supplementary Data 3). SNPs in the box plot represent the distribution of average Euclidean distances of each variable position along the MPXV genome. Box plots representing median (middle line), 25th and 75th percentile (box), and 5th and 95th percentile (whiskers) outliers are plotted as single dots; blue = LCR, yellow = SNPs. (Multiple pairwise Wilcoxon test false discovery rate [FDR] corrected *p*-value < 0.05 and specific *p*-values are indicated in the graph.).

## Conservation and variation in proteins encoded by OPGs and codon usage analysis in OPG LCRs

Analyses showed that LCRs were associated with intra-host and inter-sample variation. Although most LCRs that showed variability in our sample set (pair 1/4, 2, 3, 5, 7, pair 10/11, and 21), three (3, 7, and 21) are located in regions that, considering orthopoxvirus evolutionary history, are associated with virulence or transmission[40,41]. Noteworthy, three of the 21 highly repetitive areas identified in our intra-host variation analysis (those of LCRs 5, 6, and 7) are located in a defined central conserved region of the orthopoxvirus genome between positions 130,000 and 138,000 (Fig. 1). This region contains *OPG152* (*Cop-A25L*) (which is truncated in the MPXV genome), *OPG153* (*Cop-A26L*) (directly affected by LCR7), and *OPG154* (*Cop-A27L*). LCR7 is the only STR that is located at the center of a functional ORF. In contrast, LCR3 and LCR21 are situated in the promoter/start area, potentially modifying the ORF start site. The repeat area of LCR7 encodes a poly-D homopolymer in a nonstructured region of *OPG153* (Fig. 5a). The changes we uncovered result in the insertion of two isoleucyls. This change resembles the primary structure found in clade I viruses. In contrast, pre-2017 African subclade IIa viruses lack such insertions.

Another region of potential functional impact is the area between 170,000 to 180,000 that includes LCRs 19, 20, 2, 21 and 3 (Fig. 1). The LCR3 repeat [ACATTATAT]n is located 21 bp upstream of the putative translation start site of *OPG208* (*Cop-K2L*). Importantly, a methionyl codon is located immediately upstream of LCR3. The usage of this start codon would result in the introduction of an Ile-Ile-Tyr repeat. This codon has a medium to low probability of being used as start codon in the cognate mRNA (T base in position −3), compared to a "strong" Kozak sequence of the downstream putative start codon.

Nevertheless, LCR3 remained in-frame in all clade II MPXV samples, indicating the possibility of alternative start translation (Fig. 5b). Interestingly, LCR3 is not located in-frame in most clade I viruses. This may be significant because *OPG208* is a member of a set of genes most likely responsible for increased virulence of clade I MPXV compared to classical (pre-epidemic) subclade IIa MPXV[40]. The LCR3 tandem repeat [ACATTATAT]n is present with *n* = 52, 54, and 62 copies in epidemic subclade IIb lineage B viruses (Fig. 2a), whereas it is *n* = 7, 37, and 27 in subclade IIa MPXV isolates Sierra Leone (GenBank #AY741551), MPXV-WRAIR7-61 (GenBank #AY603973), and MPXV-COP-58 (GenBank #AY753185) sequences, respectively[40], as well as *n* = 16 for clade I MPXV isolate Zaire-96-I-16 (GenBank #AF380138). Interestingly, all publicly available subclade IIb lineage A single-molecule long-read sequence data imply a repeat *n* < 40 (Fig. 2a). The LCR3 repeat sequence may have implications beyond its repetitive nature, potentially influencing promoter function. Notably, a codon usage analysis of MPXV 2022[42] unveiled biases in the usage of different codons. This analysis also suggested an increased adaptation to primates and unique evolutionary processes within MPXV. TAC codons appear to be underrepresented in the MPXV genome[42]. LCR3 introduces 52 TAC copies. Intriguingly, when considering the alternative methionine codon upstream of LCR3, the usage of TAC codons is significantly changed compared to its regular utilization in MPXV (Fig. 5c). This altered usage pattern aligns more closely with that observed in humans (Fig. 5c), similar to the minimal adaptation to primates observed by Zhou et al.[42]. Additionally, the STR downstream of LCR21 introduces a methionyl codon upstream of the putative start codon for *OPG204* (*Cop-B19R*) (Fig. 5d). Preliminary analysis of the Kozak sequence indicates a medium to high probability for translation compared to the putative

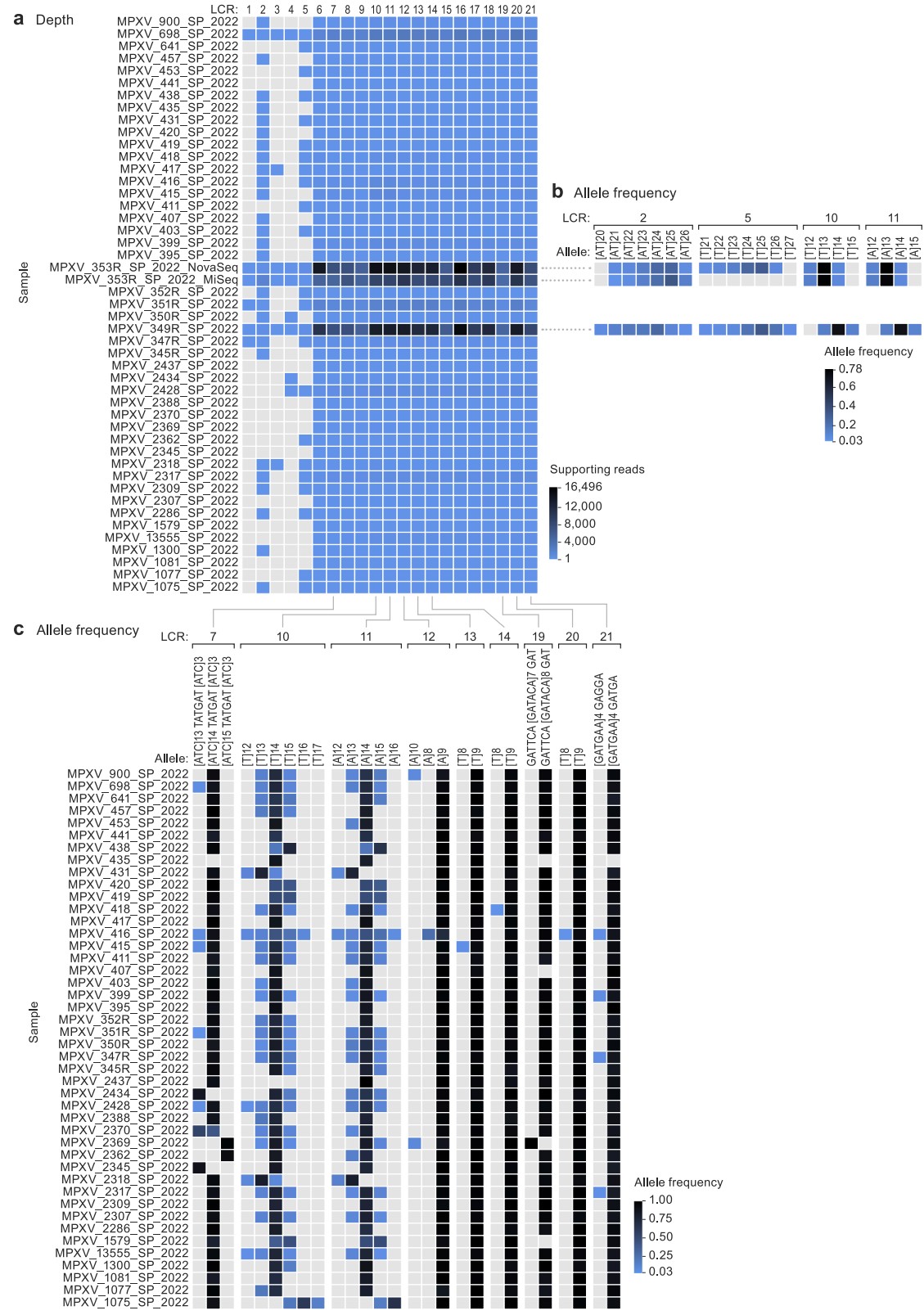

start codon (Fig. 5d). These observations highlight the nuanced influence of repeat sequences and codon usage on the translational landscape of MPXV, providing intriguing avenues for further investigation.

The remaining two LCRs (pair 4/1 and pair 10/11) are located downstream of known ORFs; thus, their variation is less likely to be associated with a change in phenotype.

## Experimental demonstration of the impact of LCR variability on viral ORF stability, transcription, and/or translation

Three LCRs are located within the putative protein-coding sections of genes *OPG153*, *OPG204*, and *OPG208*. We observed that alterations in *OPG204* and *OPG208* are found close to the N-terminal area and, thus, hypothesized they could be connected to the regulation of protein synthesis. The LCR repeat in *OPG208* gives rise to an Ile-Ile-Tyr repeat

**Fig. 4 | Low-complexity regions (LCRs) might be more phylogenetically informative than single-nucleotide polymorphisms (SNPs) for inter-host sequence analysis.** Monkeypox virus (MPXV) population genomics within the biological specimen (intra-host) and across different specimens (inter-host). **a** Panel shading indicates the number of reads supporting each LCR for each sample. Only paired reads that include a perfect match to both flanking regions were counted; the gradient shows the maximum value in black and the minimum value ($n = 1$) in the lightest blue. Samples without coverage are indicated in gray. **b** Comparison of LCR allele frequency for samples 353R and 349R. Only LCRs with at least 10 supporting paired reads including both flanking regions were counted; only alleles with a frequency of 0.03 or higher were considered. The gradient shows the maximum value in black and the minimum value ($n = 0.03$) in the lightest blue. **c** Comparison of LCR allele frequency in all samples for LCRs with good coverage (7, pair 10/11, 12, 13, 14, 19, 20, and 21). Only LCRs with at least 10 supporting paired reads including both flanking regions were counted; only alleles with a frequency of 0.03 or higher were considered. The gradient shows the maximum value in black and the minimum value ($n = 0.03$) in the lightest blue.

(Fig. 5b), and a Met-Lys repeat is created by the variation in the N-terminal domain in *OPG204* (Fig. 5d). Instead, the modification observed in *OPG153*, a poly-D amino-acid homopolymer (Fig. 5a), was located at the center of the ORF, but its resulting nature resembled features regulating protein degradation in other mammalian systems. Thus, we designed an initial experiment to demonstrate the presence of alternative start codons (ATGs) and N-terminal domains in viral mRNAs of subclade IIb MPXV strains.

### Characterization of *OPG204* and *OPG208* promoters and their transcriptomic data during MPXV infection suggest that LCRs 3 and 21 are transcribed

We utilized open-access RNA-seq data to examine alterations in the transcriptional profile of MPXV during infection[43]. We used these data to identify the top (e.g., 10 or more abundant transcripts) early and late expressed viral genes (see DESEQ results in Supplementary Data 8). Our ultimate objective was to pinpoint their conserved promoters. Both type of promoters, as previously characterized for orthopoxviruses[44,45], exhibit a high ratio of adenine and thymine bases (Figure S6a). Late promoters encompass the initial start codon of the gene, whereas early promoters are typically located approximately 20 nucleotides upstream of the start codon (Figure S6b). These characteristics are akin to those previously reported for VACV[44,45]. We successfully identified the promoter sequences of *OPG204* and *OPG208*. Interestingly, we confirmed their location upstream of the LCRs and the implied alternative start codon linked to these regions (Figure S6c and S6d).

To further substantiate the transcription of these LCRs, along with their associated alternative start codons, we assessed the read coverage of the 5′ end (spanning a region that begins 40 nucleotides upstream of the start codon and concludes 10 nucleotides downstream it) for a collection of early-expressed genes. We anticipated that areas expected to be transcribed (e.g., immediately downstream of promoters) would possess higher read coverage than regions flanking the gene (Fig. 6a–h). The LCRs and alternative start codons exhibited significantly higher read coverage than the non-coding regions (Fig. 6g and Fig. 6h). This finding held true for both *OPG204* and *OPG208*. This additional evidence supported our hypothesis that the LCRs and the ATGs associated with them are transcribed, extending the translated protein.

### Discussion

MPXV subclade IIb traces back to a human MPXV infection that likely occurred after spillover from a local animal reservoir in Ihie, Abia State, Nigeria, in 1971. An additional 10 human infections with MPXV of this lineage were detected through 1978, when this lineage seems to have disappeared. However, in 2017, it reemerged in Yenagoa, Bayelsa State, Nigeria[15]. Since then, hundreds of mpox cases have been reported and MPXV belonging to the subclade IIb lineage has been sampled in several countries. However, there were no secondary cases prior to the 2022 epidemic[46–49].

Subclade IIb viruses cause mpox that presents differently than the classical disease caused by clade I and subclade IIa viruses; i.e., subclade IIb infections are associated with higher prevalence among adults than adolescents, are predominant in the MSM community, and are efficiently transmitted human-to-human from localized infectious skin lesions rather than requiring disseminated infection[17–21].

Comparative genomics demonstrated obvious relationships between orthopoxvirus genotype and phenotype, driven by selective pressure from hosts[31,40,50–55]. Consequently, it was expected that increased MPXV genotype IIb human-to-human transmission would go hand in hand with genotypic changes. However, since orthopoxvirus genomes are organized in redundant ways[37,38,56,57], genotypic changes were expected to be modulating, rather than radical.

Genomic characterization of the 2022 epidemic has focused on describing its evolutionary history and tracking MPXV introductions into western countries. The 2022 MPXV cluster diverges from predecessor viruses by an average of 50 SNPs. Of these, the majority ($n = 24$) are non-synonymous mutations with a second minority subset of synonymous mutations ($n = 18$) and a few intergenic differences ($n = 4$)[58]. Genome editing by apolipoprotein B mRNA-editing catalytic polypeptide-like 3 (APOBEC3) and deletion of immunomodulatory genes were also noted[59,60]. MPXV sublineages mostly represent very small variations, usually characterized by one or two SNP differences to basal phylogenetic nodes[14]. Four of the determined MPXV genome sequences can be assigned to global lineage B.1.1, one to lineage B.1.3, and the remaining ones to lineage B1. Due to the limited uncovered relationship between SNPs and epidemiological links, we propose that future MPXV genomic epidemiological analyses include LCRs.

Our analyses located considerable MPXV genomic variability in areas previously considered of poor informative value, i.e., in LCRs. Because LCR entropy is significantly higher than that of SNPs and LCRs are not randomly distributed in defined coding areas in the genome and, because genomic accordions are a rapid path for orthopoxvirus adaptation during serial passaging[37,38], we posit that LCR changes might be associated with MPXV transmissibility differences over time.

Eight LCRs had evident signs of intra-host and inter-sample variation (pair 1/4, 2, 3, 5, 6, 7, pair 10/11, and 21). Five of them (5, 6, 7, 3, and 21) were co-located in two areas of the MPXV genome: base pairs 130,000–135,000 (5, 6, and 7) which are in the central conserved region of the orthopoxvirus genome in which most "housekeeping" genes are located; and base pairs 170,000–180,000 (3 and 21), which are located in the immunomodulatory area (Fig. 1). One of those LCRs is located inside the translated regions of gene *OPG153*, whereas LCRs associated with *OPG204* and *OPG208* would increase the length of the genes by extending the N-terminal domains with repetitive codons. The *OPG153* repeat results in a poly-D amino-acid homopolymer string (Fig. 5a); the LCR repeat in *OPG208* results in an Ile-Ile-Tyr repeat (Fig. 5b); and the N-terminal domain variation in *OPG204* results in a Met-Lys repeat (Fig. 5c). Thus, to demonstrate the biological relevance of our findings, we analyzed publicly available transcriptome data obtained at different times during the MPXV replication cycle, allowing for the identification of expression of viral transcripts[43]. We first identified the early and late putative MPXV promoters (Figure S6). We also demonstrated the presence of viral mRNA derived from both *OPG204* and *OPG208* that included the putative LCR-encoding region, a position of the promoter compatible with expression of the extended viral mRNA, and complete absence of the shorter previously annotated mRNA. These data demonstrate that LCR-containing areas of the genome are of functional importance.

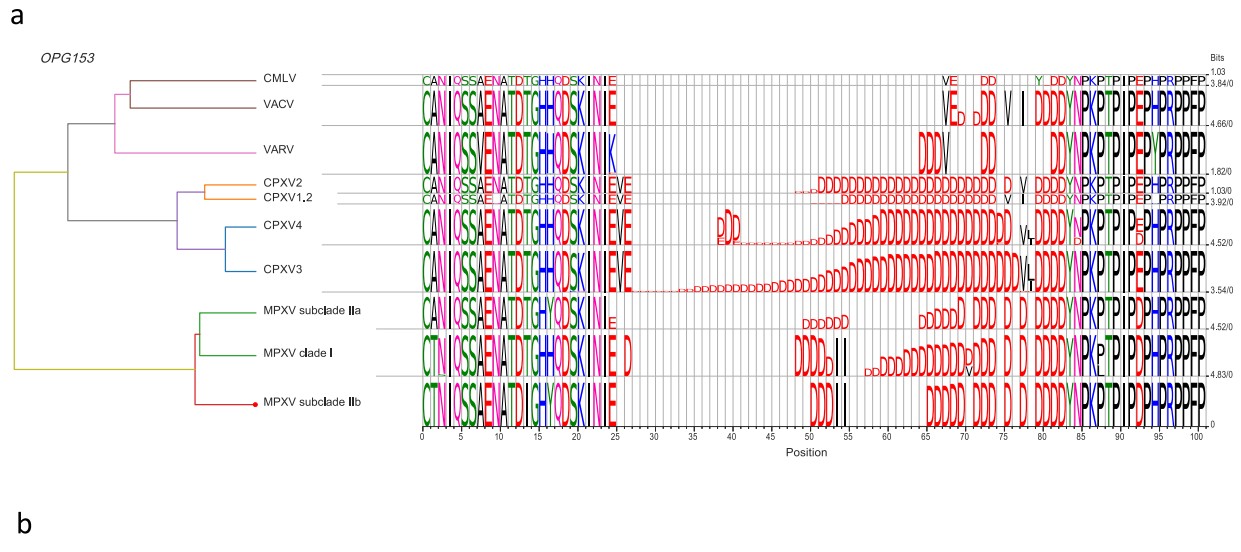

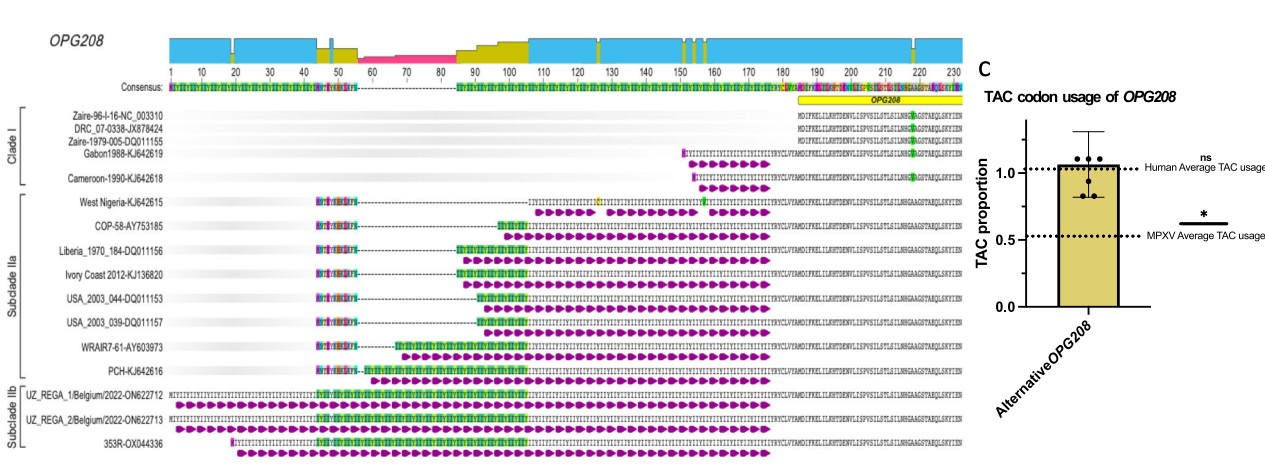

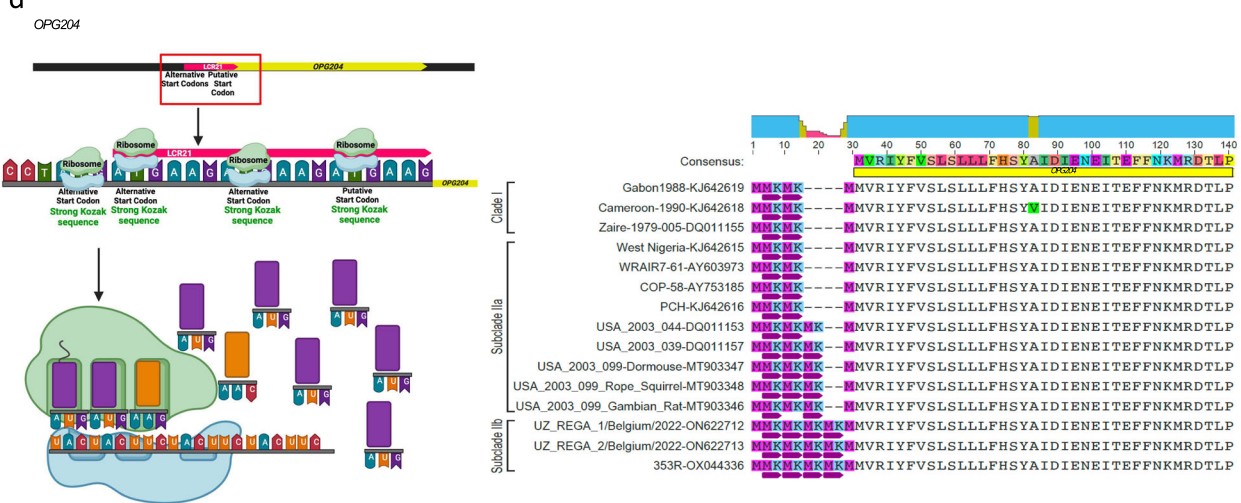

**Fig. 5 | Conservation and variation in proteins encoded by orthologous poxvirus gene (OPG) and codon usage analysis in OPG low-complexity regions (LCRs).** MetaLogo visualization of conserved and varying amino-acid residues in OPG-encoded proteins among monkeypox virus (MPXV) clade I, subclade IIa, and subclade IIb, with homologous and nonhomologous sites highlighted. Genome sequences used for each alignment can be found in Supplementary Data 5. **a** *OPG153* LCR7-derived variability. CMLV, camelpox virus; VACV, vaccinia virus; VARV, variola virus; CPXV, cowpox virus; MPXV, monkeypox virus. **b** *OPG208*

LCR3-derived variability; **c** Frequencies (mean +/− standard error) of TAC codon usage in the *OPG208* ORF using an alternative start codon (ATG) in MPXV clade IIb (*n* = 7 sequences, accession versions: OX044336.2 [generated in this study], ON563414.3, ON568298, ON736420, ON674051, ON649879, and ON676707.1) to average TAC usage in MPXV and humans (one-tailed Wilcoxon test *p*-value = 0.0156) (*=*p*-value < 0.05); and **d** *OPG204* LCR21-derived variability. Visualizations created at Biorender.com and Geneious version 2022.2 created by Biomatters.

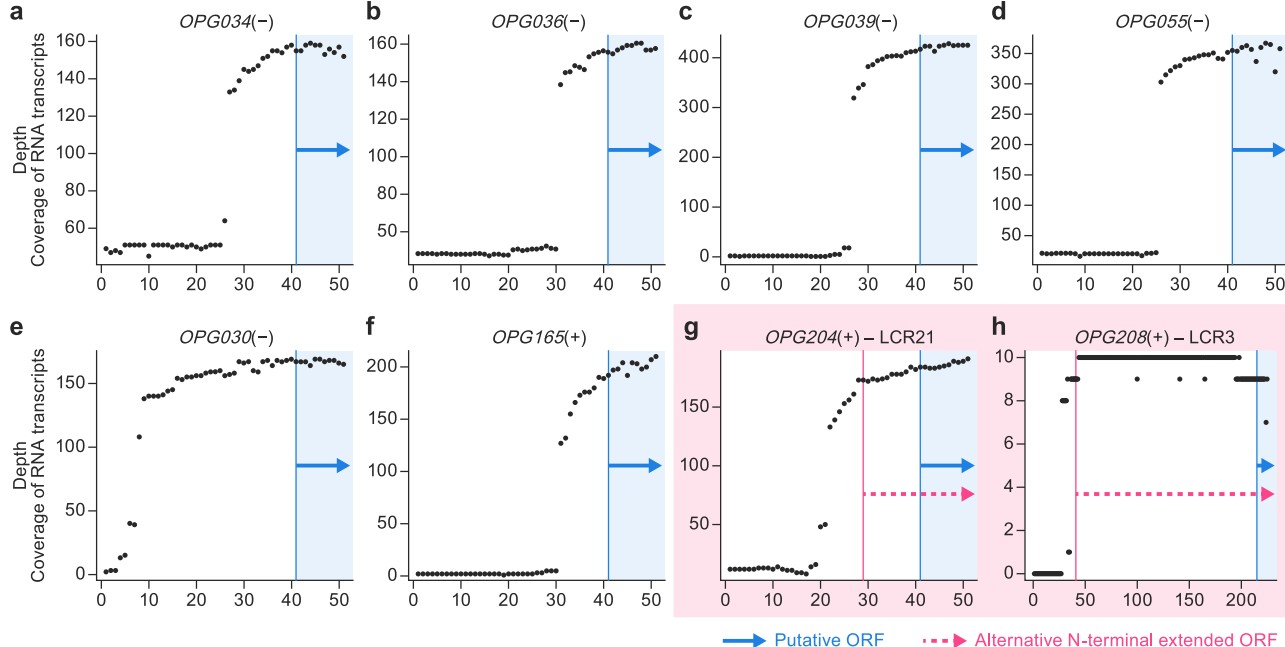

**Fig. 6 | Read coverage analyses with RNA-seq data suggest that low-complexity regions (LCRs) 3 and 21 are transcribed during monkeypox virus (MPXV) infection. a**–**g** The positional depth of translation starts, panning from the 40 nucleotides upstream to the 10 nucleotides downstream of the first alternative start codon (ATG), of top early genes. The vertical blue line represents the first ATG of the gene, and the arrow shows the spans of the first 10 (canonical) coding nucleotides. **g**, **h** If present, red lines represent ATGs, evidenced from transcriptomic data demonstrating that both *OPG204* and *OPG208* ATGs, located upstream of the canonical ones, are transcribed during MPXV subclade IIb infection. ORF, open reading frame.

Protein translation rates are in part regulated by the availability of mRNA codon-cognate amino-acylated tRNAs. The non-optimal tyrosyl codons in MPXV LCRs 3 and 21 suggest such translation modulation for *OPG208* and *OPG204*, respectively. The protein encoded by *OPG208*, Serpin-K2, is a serine protease inhibitor-like protein that functions as an inhibitor of apoptosis in VACV-infected cells[61,62] that could prevent VACV proliferation and protect nearby cells[63,64]. Our analysis points to considering *OPG208* as a potential MPXV virulence marker, opposite to what has been stated previously[40]. The protein encoded by *OPG204*, B16, is a secreted decoy receptor for interferon type I[65,66]. We did not observe any repeat number changes in *OPG204*-associated LCR21, but SNPs in clade I, subclade IIa, and subclade IIb result in alternative translational start sites followed by a lysyl, suggesting these SNPs could have direct effects on *OPG204* translation.

The changes observed in *OPG153* stand out as they are located inside a region that is under high selective pressure for transmission in a "housekeeping" orthologous poxvirus gene, which is involved in virion attachment and egress[32]. Thus, we urge that these areas be scrutinized for changes that might affect the MPXV interactome.

The *OPG153* expression product (A26) attaches orthopoxvirus particles to laminin[67] and regulates their egress[41,68–70]. *OPG153* is unique, as it is the central conserved region gene that has been "lost" the most times during orthopoxvirus evolution[32]. Inactivation of *OPG153* genes by frameshift mutations occurs rapidly in experimental evolution models[38], resulting in increased virus replication levels, changes in particle morphogenesis, decreased particle-to-PFU ratios, and pathogenesis modulation[38,41]. Finally, A26 is the main target of the host antibody response to orthopoxvirus infection[71,72]. Thus, any *OPG153* modulation is likely of significance.

In summary, our in silico comparative genomics and in vitro functional genomics findings expand the concept of genome accordions as a simple and recurrent mechanism of adaptation on a genomic scale in orthopoxvirus evolution. A consequence of this broadening is the recognition that areas of the MPXV genome (LCRs) might be relevant in adaptability of the orthopoxvirus replication cycle. Further characterization of their functional biology, which will need to incorporate "loss-of-function" and genetically modified strains to understand their roles, will likely improve our understanding of current mpox epidemiology and clinical presentation.

## Methods
### Study population
This study includes confirmed human mpox cases diagnosed from May 18 to July 14, 2022, at the Centro Nacional de Microbiología (CNM), Instituto de Salud Carlos III, Madrid, Spain. The study was performed as part of the public health response to the current mpox epidemic by the Spanish Ministry of Health. Sample information is listed in Supplementary Data 1 and Supplementary Data 5.

The samples used in this work were obtained in the context of the Microbiological Surveillance and Diagnosis Program for the mpox outbreak conducted by the Centro Nacional de Microbiología, Instituto de Salud Carlos III. The study was based on routine testing, did not involve any additional sampling or tests and stored RNA extracts were used, so specific ethical approval was not required for this study. All sequenced viruses corresponded to those to patients that gave consent to be analyzed for diagnosis or surveillance purposes.

### Study sample processing
Swabs of vesicular lesions from study patients in viral transport media were sent refrigerated to CNM. Nucleic acids were extracted at CNM using either a QIAamp MinElute Virus Spin DNA Kit (#57704; Qiagen, Germantown, MD, USA) or QIAamp Viral RNA Mini Kit (#52904; Qiagen) according to the manufacturer's recommendations. Inactivation of samples was conducted in a certified class II biological safety cabinet in a biosafety level (BSL) 2 laboratory using BSL-3 best practices with appropriate personal protective equipment.

### MPXV laboratory confirmation
MPXV detection by PCR in a sample was considered laboratory confirmation and resulted in inclusion of the swab in the study. A

previously described orthopoxvirus-generic real-time PCR (qPCR) was used for screening[73]. A previously described conventional validated nested PCR targeting *OPG002* (*Cop-C22L*) (encoding a TFN receptor) was used for results confirmation[74].

## MPXV genome sequencing

Sequencing libraries were prepared with a tagmentation-based Illumina DNA Prep kit (#20060060; Illumina, San Diego, CA, USA) and run in a NovaSeq 6000 SP Reagent Kit (#20028312; Illumina) flow cell using 2×150 paired-end sequencing. To improve assembly quality, the library from swab 353R, an unpassaged vesicular fluid from a confirmed case, was also run in a MiSeq Reagent Kit v3 (#MS-102-3003; Illumina) flow cell using 2×300 paired-end sequencing. Additionally, sample 353R was also analyzed by single-molecule methods using nanopore sequencing (Oxford Nanopore Technologies, Oxford, UK). For nanopore sequencing, 210 ng of DNA was extracted from swab 353R and used to prepare a sequence library with a Rapid Sequencing Kit (#SQK-RAD114; Oxford Nanopore Technologies); the library was analyzed in an FLO-MIN106D (#FLO-MIN106D; Oxford Nanopore Technologies) flow cell for 25 h. The process rendered 1.12 Gb of filter-passed bases.

## Bioinformatics

**De novo assembly and annotation of subclade IIb lineage B.1 MPXV genome sequence 353R.** Due to the high yield of MPXV genomic material in a preparatory run, swab 353R was selected as source material for the determination of an MPXV high-quality genome (HQG) sequence. Single-molecule long-sequencing reads were preprocessed using Porechop v0.3.2pre[75] with default parameters. Reads were de novo assembled using Flye v2.9-b1768[76] in single-molecule sequencing raw read mode with default parameters, resulting in one MPXV contig of 198,254 bp. Short 2×150 sequencing reads were mapped with Bowtie2 v2.4.4[77] against the selected contig, and resulting BAM files were used to correct the assembly using Pilon v1.24[78]. At this intermediate step, this corrected sequence was used as a reference in the nf-core/viralrecon v2.4.1 pipeline[79] for mapping and consensus generation with short-read sequencing. The allele frequency threshold of 0.5 was used for including variant positions in the corrected contig.

Short MiSeq 2×300 and NovaSeq 2×150 sequencing reads were also assembled de novo using the nf-core/viralrecon v2.4.1 pipeline, written in Nextflow[80] in collaboration between the nf-core community[81] and the Unidad de Bioinformática, Instituto de Salud Carlos III, Madrid, Spain (https://github.com/BU-ISCIII). FASTQ files containing raw reads were quality controlled using FASTQC v0.11.9[82]. Raw reads were trimmed using fastp v0.23.2[83]. The sliding-window quality-filtering approach was performed, scanning the read with a 4-base-wide sliding window and cutting 3' and 5' base ends when average quality per base dropped below a Qphred of 20. Reads shorter than 50 bp and reads with more than 10% read quality under Qphred 20 were removed. Host genome reads were removed via kmer-based mapping of the trimmed reads against the human genome reference sequence GRCh38 (https://www.ncbi.nlm.nih.gov/data-hub/genome/GCF_000001405.26/) using Kraken 2 v2.1.2[84]. The remaining non-host reads were assembled using SPADES v3.15.3[85,86] in rnaviral mode. A fully ordered MPXV genome sequence was generated using ABACAS v1.3.1[87], based on the MPXV isolate MPXV_USA_2022_MA001 (Nextstrain subclade IIb lineage B.1) sequence (GenBank #ON563414.3)[88]. The independently obtained de novo assemblies and reference-based consensus genomes obtained from swab 353R were aligned using MAFFT v7.475[89] and visually inspected for variation using Jalview v2.11.0[90].

**Systematic identification of low-complexity regions in orthopoxvirus genomes.** Detection of STRs in the HQG sequence and other orthopoxvirus genomes was performed with Tandem repeats finder[91], using default parameters. Briefly, the algorithm works without the need to specify either the pattern or its length. Tandem repeats are identified considering percent identity and frequency of insertion (ins) or deletion (del) of bases (indels) between adjacent pattern copies, using statistically based recognition criteria. Since Tandem repeats finder does not detect single-nucleotide repeats, we developed an R script to systematically identify homopolymers of at least 9 nucleotide residues in all available orthopoxvirus genome sequences. STRs and homopolymers were annotated as LCRs.

**Curation of LCRs in the MPXV high-quality virus genome sequence.** We curated LCRs in the HQG sequence using a modified version of STRsearch[92]. Once provided with identifying flanking regions, STRsearch performed a profile analysis of STRs in massively parallel sequencing data. To ensure high-quality characterization of LCR alleles, we modified the script (https://github.com/BU-ISCIII/MPXstreveal) to complement reverse reads that map against the reverse genome strand according to their BAM flags. In addition, output was modified to add information later accessed by a custom Python script to select only reads containing both LCR flanking regions. All LCRs in the HQG sequence were manually validated using STRsearch results and de novo assemblies obtained from all sequencing approaches. When an LCR was only resolved by single-molecule long-sequencing technologies (LCR pair 1/4 and LCR3), we also analyzed publicly available data by downloading all single-molecule long-sequencing data from NCBI SRA (https://www.ncbi.nlm.nih.gov/sra) as of August 10, 2022, and analyzed the data according to Supplementary Note 1 found in Supplementary Information file.

**Final MPXV high-quality virus genome sequence assembly.** The consensus genome constructed with the nf-core/viralrecon v2.4.1 pipeline using the corrected de novo contig as stated above, along with the resulting curated and validated consensus LCRs, were used to build the final HQG reference sequence using a custom Python script. The resulting HQG is available from the European Nucleotide Archive (#OX044336.2).

**Generation of MPXV high-quality virus genome reference-based consensus sequence for all other samples.** For the remaining specimens, sequencing reads were analyzed for MPXV genome sequence determination using the nf-core/viralrecon v2.4.1 pipeline. Trimmed reads were mapped with Bowtie2 v2.4.4 against the HQG sequence and the sequence of subclade IIb lineage A MPXV isolate M5312_HM12_Rivers (GenBank #MT903340.1)[93]. Picard v2.26.10[94] and SAMtools v1.14[95] were used to generate MPXV genome mapping statistics. iVar v1.3.1[96], which calls for low-frequency and high-frequency variants, was used for variant calling. Variants with an allele frequency higher than 75% were kept to be included in the consensus genome sequence. BCFtools v1.14[97] was used to obtain the MPXV genome sequence consensus with filtered variants and masked genomic regions with coverage values lower than 10X. All variants, included or not, in the consensus genome sequence, were annotated using SnpEff v5.0e[98] and SnpSift v4.3[99]. Final summary reports were created using MultiQC v1.11[100]. Consensus genome sequences were analyzed with Nextclade v2.4.1[101] using the "MPXV (All clades)" dataset (timestamp 2022-08-19T12:00:00Z). Raw reads and consensus genomes are available from the European Nucleotide Archive (#ERS12168855–ERS12168865, #ERS12168867, #ERS12168868, and #ERS13490510–ERS13490543).

**Intra-host and inter-host allele frequency analyses.** Intra-host genetic entropy (defined as -sum(Xi*log(Xi)), in which Xi denotes each of the allele frequencies in a position) was calculated according to the SNP frequencies of each position along the genome using nf-core/viralrecon v2.4.1 pipeline results. Similarly, genetic entropy for each LCR was calculated considering the frequencies of repeat lengths.

LCR intra-host and inter-host variations in the sample set were analyzed using a modified version of STRsearch. As a quality check for this analysis, STRsearch results (Supplementary Data 5) were filtered, keeping alleles with at least 10 reads spanning the region and allele frequency above 0.03. Quality control and allele frequency graphs were created using a customized R script.

Pairwise genetic distances between samples were calculated as Euclidean distances (defined as /X-Y/=sqrt(sum(xi-yi)^2), in which xi and yi are the allele frequencies of sample x and y at a given position, respectively), thus accounting for the major and minor alleles at each analyzed position. Distances were calculated individually for each variable LCR (STRs 2, 5, 7, 10, 11, and 21) and for each of all 5422 SNPs showing inter-sample variability (compared to MPXV-M5312_HM12_Rivers). The distributions of inter-sample distances were compared between LCRs using a Kruskal–Wallis test ($\chi^2$ $p$-values) followed by multiple pairwise-comparison between groups (Wilcoxon test), with $p$-values subjected to the false discovery rate (FDR) correction. A randomization test was used to test whether inter-sample variability in LCRs is higher than that in SNPs: first, the average Euclidean distance for each LCR and each SNP position was calculated; then, the average value of each LCR was compared to a random sample of 1000 values from the distribution of mean distances from the SNPs along the genome. The $p$-value was calculated from the percentage of times that the mean of the LCR was higher than the randomly taken values from the SNPs.

**Phylogenetic analysis of the MPXV central conserved region.** Variant calling and SNP matrix generation was performed using Snippy v4.4.5[102], including sequence samples and representative MPXV genome sequences downloaded from GenBank (Supplementary Data 5). The SNP matrix with both invariant and variant sites was used for phylogenetic analysis using IQ-Tree 2 v. 2.1.4-beta[103] via predicted model K3Pu+F + I and 1000 bootstrap replicates. A phylogenetic tree was visualized and annotated using iTOL v6.5.8[104]. The SNP matrix was also used for generating the haplotype network using PopArt v1.7[105].

**Selected MPXV ORF analysis.** Representative orthopoxvirus genomes[32] were downloaded from GenBank together with the consensus genome sequences from the specimens analyzed in this study (Supplementary Data 5). MPXV genomes were assigned to clades and lineages according to the most recent nomenclature recommendations according to Nextstrain[14] using Nextclade v2.4.1. Annotations from RefSeq #NC_063383.1 (subclade IIb lineage A MPXV virus isolate MPXV-M5312_HM12_Rivers) GFF file were transferred to all FASTA genome sequences using Liftoff v1.6.3[106]. *OPG153* was extracted using AGAT v0.9.1[107] and multi-FASTA files were generated for each group and gene. *OPG204* and *OPG208* alternative annotation start site ORFs were re-annotated in Geneious Prime (Biomatters, San Diego, CA, USA), and extracted as alignments. We used MUSCLE v3.8.1551 for aligning each multi-FASTA file and Jalview v2.11.0 for inspecting and editing the alignments. Finally, MetaLogo v1.1.2[108] was used for creating and aligning the sequence logos for each orthopoxvirus group of the *OPG153* LCR7, *OPG204* LCR21, and *OPG208* LCR3.

**Comparison of LCR frequencies in protein functional groups.** The potential biological impact of LCRs was evaluated by mapping the frequency and location of STRs and homopolymers in the orthopoxvirus genome and considering the biological function of the affected genes. The frequency of inclusion of LCRs between distinct functional groups of genes was compared as previously described[32]. Orthopoxvirus genomes ($n$ = 231, Akhmeta virus [AKMV]: $n$ = 6 sequences; alaskapox virus [AKPV]: $n$ = 1; cowpox virus [CPXV]: $n$ = 82; ectromelia virus [ECTV]: $n$ = 5; MPXV: $n$ = 62; VACV: $n$ = 18; VARV: $n$ = 57) include 216 functionally annotated OPGs classified in 6 categories ("Housekeeping genes/Core" ANK/PRANC family, Bcl-2 domain family, BTB/Kelch domain family, PIE

family, and "Accessory/Other" [e.g., virus–host interacting genes]). The frequency was calculated after normalizing the number of LCRs registered with the sample size of the OPG alignment in the categories described above. Statistical analysis of the significance of differences was performed by applying a Kruskal–Wallis test ($\chi^2$ $p$-values) followed by a non-parametric multiple pairwise comparison between groups (Wilcoxon test), with $p$-values subjected to FDR correction.

**Viral Transcriptomics**

**Data selection.** RNA-seq samples were retrieved from NCBI SRA (ncbi.nlm.nih.gov/sra) under accession ERP141806. The RNA-seq data are from MPXV isolated from skin lesion samples from a patient; samples had been cultured in CV-1 cells for 1–24 h post-infection, 3 replicates each. The original samples were subjected to total RNA isolation, library preparation, poly(A)+ enrichment, and RNA sequencing with ONT MinION. The BAM files retrieved from SRA were preprocessed files from which reads have been aligned to MPXV ON563414.3 and human GCF_015252025.1[43].

**DEG analyses.** Default parameters were set in htseq-count[109], which was used to obtain transcript counts for each MPXV gene (ON563414.3 reference) for every sample. Differential gene-expression analyses were then performed with DESeq2 (R package)[110] to differentiate early (those overexpressed at early time points) versus late (most expressed at late time points) viral genes. A linear-regression analysis considering the variables "replicate" and "timepoint" was performed.

**Definition of promoters.** For the top 10 early and late genes (defined as those genes with padj <0.05 that had the largest absolute log2 fold change values), we retrieved the sequence representing the 100 nucleotides upstream of their canonical start codon (ATG), considering the coordinates of the reference genome ON563414.3. Early and late genes were then aligned separately with the Muscle alignment tool[111] from MEGA11[112]. VACV early and late genes have unique conserved TATA boxes[44,45,113]. We used these reports as templates to define the likely promoters in MPXV.

After defining the more likely consensus early and late promoters from the general viral transcripts, we identified the specific promoters of *OPG204* (which was among the top upregulated genes at 1 h post-infection) and of *OPG208* (not overexpressed at 1 nor at 24 h post-infection but expressed at the earliest timepoint).

**Read depth analysis.** We later analyzed the region spanning 40 bases before the start of translation to the first 10 nucleotides from the analyzed coding region of all top early genes using SAMtools[95] to assess the read depth at the 5' end, and later we compared them to *OPG204* and *OPG208*, which were both defined as early genes. The idea behind this analysis is that, given poly(A)+ enrichment, regions that are transcribed (e.g., containing a coding region and untranslated 5' region) should be covered at significantly higher depth than non-transcribed areas (noise).

**Reporting summary**
Further information on research design is available in the Nature Portfolio Reporting Summary linked to this article.

## Data availability

The data generated in this study are provided within the article, Supplementary Information, and Supplementary Data files. MPXV high-quality genome sequence is available in the European Nucleotide Archive (ENA) database (Accession number OX044336) https://www.ebi.ac.uk/ena/browser/view/OX044336, accession numbers for all the samples sequenced in this manuscript can be found in Supplementary Data 1. Further information and requests for resources and reagents should be directed to and will be fulfilled by the lead contact, Gustavo Palacios (gustavo.palacios@mssm.edu).

## Code availability

All scripts and codes used for this study can be found at GitHub repository: (https://github.com/BU-ISCIII/MPXstreveal)[114].

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

## Acknowledgements

We would like to thank the work of the Rapid Response Unit of the National Center for Microbiology, especially MªJosé Buitrago, and Cristobal Belda, ISCIII General Director. We also thank Anya Crane (Integrated Research Facility at Fort Detrick, National Institute of Allergy and Infectious Diseases, National Institutes of Health) for critically editing the manuscript and Jiro Wada (Integrated Research Facility at Fort Detrick, National Institute of Allergy and Infectious Diseases, National Institutes of Health) for helping with figure preparation. The work for this study performed at Instituto de Salud Carlos III was partially funded by Acción Estratégica "Impacto clínico y microbiológico del brote por el virus de la viruela del mono en pacientes en España (2022): proyecto multicéntrico MONKPOX-ESP22" (CIBERINFEC) (M.P.S.S.). The work for this study done at the Icahn School of Medicine at Mount Sinai Department of Microbiology as part of Global Health Emerging Pathogen Institute activities was funded by institutional funds (G.P.) from the Icahn School of Medicine at Mount Sinai Department of Microbiology in support of Global Health Emerging Pathogen Institute activities. This work was also supported in part through Laulima Government Solutions, LLC, prime contract with the U.S. National Institute of Allergy and Infectious Diseases (NIAID) under Contract No. HHSN272201800013C. J.H.K. performed this work as an employee of Tunnell Government Services (TGS), a subcontractor of Laulima Government Solutions, LLC, under Contract No. HHSN272201800013C. Opinions, interpretations, conclusions, and recommendations are those of the authors and are not necessarily endorsed by the U.S. Army. The views and conclusions contained in this document are those of the authors and should not be interpreted as necessarily representing the official policies, either expressed or implied, of the U.S. Department of Health and Human Services or of the institutions and companies affiliated with the authors, nor does mention of trade names, commercial products, or organizations imply endorsement by the U.S. Government.

## Author contributions

Conceptualization, S.M., S.V., A.N., I.J., M.S.-L., A.G.-S., G.P., I.C., and M.P.S.S. Methodology, S.M., S.V., A.N., S.V.-F., J.A.P.-G., N.F.-G., A.Z., J.H.K., M.S.-L., N.D., J.R.K., E.G., S.G., and G.P. Investigation, S.M., S.V., A.N., S.V.-F., J.A.P.-G., N.F.-G.,A.Z., E.O., O.A., A.M.-G., A.D.-I., V.E., C.G., F.M., P.S.-M., M.T., A.V., J.-C.G., I.T., M.C. del R., L.M-D., M.L., A.G., L.C., A.G., J.C., L.H., P.J., M.L.N.-R., I.J., M.E.A.-A., C.L., L. del R., I.E., M.S., M.A.M., J.H.K., M.S.-L., N.D.P., J.R.K., E.G., S.G., A.G.-S., G.P., I.C., and M.P.S.S. Formal Analysis, S.M., S.V., A.N., S.V.-F., J.A.P.-G., N.F.-G., J.H.K., M.S.-L., N.D.P., J.R.K., E.G., G.P., I.C., and M.S.S. Writing – Original Draft, S.M., S.V., A.N., S.V.-F., and G.P. Writing – Review & Editing, S.M., S.V., A.N., S.V.-F., J.H.K., A.G.-S., G.P., I.C., and M.P.S.S. Visualization, S.M., S.V., A.N., S.V.-F., and G.P. Supervision, A.G.-S., G.P., I.C., and M.P.S.S. Resources, A.G.-S., G.P., I.C., and M.P.S.S. Funding Acquisition, A.G.-S., G.P., I.C., and M.P.S.S.

## Competing interests

A.G.-S. has consulting agreements for the following companies involving cash and/or stock: Castlevax, Amovir, Vivaldi Biosciences, Contrafect, 7Hills Pharma, Avimex, Vaxalto, Pagoda, Accurius, Esperovax, Farmak, Applied Biological Laboratories, Pharmamar, Paratus, CureLab Oncology, CureLab Veterinary, Synairgen, and Pfizer, outside of the reported work. A.G.-S. has been an invited speaker in meeting events organized by Seqirus, Janssen, Abbott, and Astrazeneca. A.G.-S. is inventor on patents and patent applications on the use of antivirals and vaccines for the treatment and prevention of virus infections and cancer, owned by the Icahn School of Medicine at Mount Sinai, New York,

outside of the reported work. The other authors declare no competing interests.

## Additional information

Sara Monzón[1,30], Sarai Varona[1,2,30], Anabel Negredo[3,4,30], Santiago Vidal-Freire[5,30], Juan Angel Patiño-Galindo[5], Natalia Ferressini-Gerpe[5], Angel Zaballos[6], Eva Orviz[7], Oskar Ayerdi[7], Ana Muñoz-Gómez[7], Alberto Delgado-Iribarren[7], Vicente Estrada[4,7], Cristina García[3], Francisca Molero[3], Patricia Sánchez-Mora[3,4], Montserrat Torres[3,4], Ana Vázquez[3,8], Juan-Carlos Galán[8,9], Ignacio Torres[10], Manuel Causse del Río[11], Laura Merino-Diaz[12], Marcos López[13], Alicia Galar[14], Laura Cardeñoso[15], Almudena Gutiérrez[16], Cristina Loras[17], Isabel Escribano[18], Marta E. Alvarez-Argüelles[19], Leticia del Río[20], María Simón[21], María Angeles Meléndez[22], Juan Camacho[3], Laura Herrero[3,4], Pilar Jiménez[6], María Luisa Navarro-Rico[6], Isabel Jado[3], Elaina Giannetti[5], Jens H. Kuhn[23], Mariano Sanchez-Lockhart[24], Nicholas Di Paola[24], Jeffrey R. Kugelman[24], Susana Guerra[5,25,26], Adolfo García-Sastre[5,25,27,28,29], Isabel Cuesta[1,31], Maripaz P. Sánchez-Seco[3,4,31] & Gustavo Palacios[5,25,31] ✉

[1]Unidad de Bioinformática, Unidades Centrales Científico Técnicas, Instituto de Salud Carlos III, 28029 Madrid, Spain. [2]Escuela Internacional de Doctorado de la UNED (EIDUNED), Universidad Nacional de Educación a Distancia (UNED), 2832 Madrid, Spain. [3]Centro Nacional de Microbiología, Instituto de Salud Carlos III, 28029 Madrid, Spain. [4]Centro de Investigación Biomédica en Red de Enfermedades Infecciosas (CIBERINFEC), Instituto de Salud Carlos III, 28029 Madrid, Spain. [5]Department of Microbiology, Icahn School of Medicine at Mount Sinai, New York, NY 10029, USA. [6]Unidad de Genómica, Unidades Centrales Científico Técnicas, Instituto de Salud Carlos III, 28029 Madrid, Spain. [7]Centro Sanitario Sandoval, Hospital Clínico San Carlos, 28040 Madrid, Spain. [8]Centro de Investigación Biomédica en Red de Epidemiología y Salud Pública (CIBERESP), Instituto de Salud Carlos III, 28029 Madrid, Spain. [9]Servicio de Microbiología, Hospital Universitario Ramón y Cajal, Instituto Ramón y Cajal de Investigación Sanitaria (IRYCIS), 28034 Madrid, Spain. [10]Servicio de Microbiología, Hospital Clínico Universitario, Instituto de Investigación INCLIVA, 46010 Valencia, Spain. [11]Unidad de Microbiología, Hospital Universitario Reina Sofía, Instituto Maimónides de Investigación Biomédica de Córdoba, 14004 Córdoba, Spain. [12]Unidad Clínico de Enfermedades Infecciosas, Microbiología y Medicina Preventiva, Hospital Universitario Virgen del Rocío, 41013 Sevilla, Spain. [13]Servicio de Microbiología y Parasitología, Hospital Universitario Puerta de Hierro Majadahonda, 28222 Madrid, Spain. [14]Servicio de Microbiología Clínica y Enfermedades Infecciosas, Hospital General Universitario Gregorio Marañón, 28007 Madrid, Spain. [15]Servicio de Microbiología, Instituto de Investigación Sanitaria, Hospital Universitario de la Princesa, 28006 Madrid, Spain. [16]Servicio de Microbiología y Parasitología Clínica, Hospital Universitario La Paz, 28046 Madrid, Spain. [17]Servicio de Microbiología, Hospital General y Universitario, 13005 Ciudad Real, Spain. [18]Servicio de Microbiología, Hospital General Universitario Dr. Balmis, 03010 Alicante, Spain. [19]Servicio de Microbiología, Hospital Universitario Central de Asturias, 33006 Asturias, Spain. [20]Hospital Quironsalud Torrevieja, 03184 Alicante, Spain. [21]Servicio de Microbiología, Hospital Central de la Defensa "Gómez Ulla", 28947 Madrid, Spain. [22]Servicio de Microbiología y Parasitología, Hospital Universitario 12 de Octubre, 28041 Madrid, Spain. [23]Integrated Research Facility at Fort Detrick, National Institute of Allergy and Infectious Diseases, National Institutes of Health, Fort Detrick, Frederick, MD 21702, USA. [24]United States Army Research Institute for Infectious Disease, Fort Detrick, Frederick, MD 21702, USA. [25]Global Health Emerging Pathogens Institute, Icahn School of Medicine at Mount Sinai, New York, NY 10029, USA. [26]Departmento de Medicina Preventiva, Salud Publica y Microbiología, Universidad Autónoma de Madrid, 28029 Madrid, Spain. [27]Department of Medicine, Division of Infectious Diseases, Icahn School of Medicine at Mount Sinai, New York, NY 10029, USA. [28]The Tisch Cancer Institute, Icahn School of Medicine at Mount Sinai, New York, NY 10029, USA. [29]Department of Pathology, Molecular and Cell-Based Medicine, Icahn School of Medicine at Mount Sinai, New York, NY 10029, USA. [30]These authors contributed equally: Sara Monzón, Sarai Varona, Anabel Negredo, Santiago Vidal-Freire. [31]These authors jointly supervised this work: Isabel Cuesta, Maripaz P. Sánchez-Seco, Gustavo Palacios. ✉e-mail: gustavo.palacios@mssm.edu

