## [Peer Review File · Nature Communications]

Reviewers' Comments:

Reviewer #1:

Remarks to the Author:

The authors present an interesting analysis of LCR diversity in MPXV genomes. They use an improved approach to resolve LCRs and they study their distribution and possible functional effects. Most analyses are sound and accurate. However, as mentioned below, the experimental part is weak and largely preliminary. Additional experiments are needed to support their conclusions. As a consequence, the overall message should be down-toned (in all sections) and a more balanced discussion should be provided. This also includes acknowledging that forces such as drift and relaxation of selective constraint have an impact on genetic diversity, including at LCRs. It would be good to read a "limitations" section in the discussion.

Lines 133 to 137 and 145-148: please add references.

Lines 151-154: can these observations be interpreted as weakening of selective constraint? Is it possible that genetic variability in those regions simply indicates that they are non-essential or, in general, less constrained? To me this is a sensible hypothesis.

Lines 168-172: the authors fail to account for other possible forces, namely relaxation of selective pressure and drift.

Lines 296-310: the higher diversity in LCRs than in SNPs is not surprising and was observed in several other systems. LCR diversity is often thought to result from polymerase slippage or other replication errors.

Lines 309-310: the last sentence should be rephrased (what is phenotypic MPXV?). Also, I don't see a reason why studies should refocus from SNPs to LCRs. Maybe they should also include LCRs, but the sheer number of differences does not necessarily imply that one type of variation is phenotypically more important than another.

Lines 381-383: if they want to speculate on codon usage, they should perform a thorough analysis-it is well known that viruses often have codon usage preferences that do not match those of their hosts.

Lines 430-442. An appropriate control should be used in the luciferase assay. If the authors want to show that LCR21 impairs protein expression, a sequence of the same length should be inserted in the NanoLuc vector as a control. The sequence can be a scrambled version of LCR21 or an unrelated sequence. Also, experiments should be performed to determine whether the impairment occurs at the transcriptional or translational level.

Lines 446-447. Please describe pDEST-CMV-N... and explain why it was used and to which purpose.

Lines 452-453. Which CPXV sequence are they referring to? As also shown in fig 5, CPXV is genetically diverse.

Lines 452-462. The results of these experiments are very preliminary and provide limited evidence for a role of LCR7 in the regulation of "degradation activity" (whatever this means).

Also, the conclusion regarding OPG153 from different MPXV clades seems to be at odds with their hypothesis: if clade I has the longest repeat, why is IIb recovered with MG-132 treatment?

Lines 489-496: this paragraph is really unclear and should be rephrased.

Lines 498-503: the conclusion of this extremely long sentence (that changes in LCRs are associated with different transmissibility) is not supported by their data.

Lines 521-522: the data do not demonstrate "unequivocally" that the LCRs have functional importance. For the reasons mentioned above, the presented data do not allow any definitive conclusion about the functional effects of LCR7 and LCR21. Even if the experiments were extended (to include appropriate controls), the "importance" of the functional effects cannot be assessed without infection assays.

Lines 523-526: they do not demonstrate that the effect is at the translational level.

Lines 553-568: this section on homopolymers in other systems is too long and useless within the framework of this manuscript.

Lines 586-594: besides the fact that it is unclear which critics the authors are referring to, it is my opinion that their data provide no evidence of selective pressure acting on LCRs (the "accordion" might be simply due to increased replication errors in low complexity regions with limited effect on fitness). In general, their conclusions should be down-toned especially with regard to the functionality of the LCR variation.

Reviewer #2:

Remarks to the Author:

In this manuscript, Monzon et al. sequence 48 Monkeypox virus (MPXV) isolates from patients from the 2022 outbreak and by applying several sequencing approaches to one high-quality isolate, they provide a high resolution genome sequence. By then comparing these with existing single genome sequences, the authors provide a more refined understanding of variability within a specific type of repeat sequence known as low complexity regions (LCRs), which vary and are claimed to be of functional importance to the pathogenesis of the current outbreak. Beyond providing a higher resolution sequence of the Clade IIb MPXV that is associated with the ongoing outbreak, the core findings that LCRs are variable in length and have potential functions are not novel. Functional testing of the three primary LCRs identified is poorly controlled and limited in scope, and seems like a rushed addition in response to a prior review. Overall, there is limited new information in this manuscript beyond sequencing through LCRs while the core claims in the title and throughout the paper are completely unsubstantiated.

Main Concerns

There are a large number of MPXV sequences available and although this one is perhaps the highest resolution to date, the main discovery is that LCRs vary in length. This is already known and to claim that these regions are a new form of gene accordion seems unjustified. These regions simply vary and this is well known, and the authors show no functional consequence of this variation.

In the title and throughout the paper, the authors also claim these sequences may open up the "pallets" to increased MPXV transmission. I am not sure what the authors mean by "pallets" but they provide absolutely no evidence of links to altered virus protein production or replication, let alone transmissibility. While many studies are trying to find a genetic link to the increase in MPXV spread during the current outbreak, it is also quite likely that there is no specific genetic basis and instead this may be a simple case of MPXV having been introduced to the MSM community where it is now spreading by a new route of sexual transmission, rather than animal bites etc.

While the sequencing side of this report is adequate, many of the identified LCR sequences are in regions of unlikely consequence and variability is to be expected. For the three that are focused upon as potentially having a functional consequence, based on the discussion it seems that the very limited

functional data was hurriedly added to figure 7 in response to prior reviews. However, the experiments are lacking in controls and underdeveloped, and do not support the claims of translational and protein stability control, let alone roles in virus transmission. For LCR21 in figure 7a-b, the control construct should have a UTR of the same length, otherwise this could simply reflect the well-established effect of long or structured versus short or unstructured UTRs that is being examined, rather than anything specific to these sequences. I would also guess that any UTR containing multiple ATG start sites would affect translation in a reporter assay but this does not mean this occurs during infection. There is also no measurement of RNA levels and all that is measured is luciferase activity, which could reflect transcription, translation, protein folding or stability. Plus, this is all done by transfection and outside the relevant context of infection. The constructs use a CMV promoter rather than the native viral promoters, which seems important to test. For OPG153 in figure 7d-e, the results seem very strange and artificial. First off, none of the constructs express detectable protein unless cells are treated with MG132. This means that the variable poly-D homopolymer sequences have no detectable impact on expression in a normal context. None of the experiments measure mRNA levels to ensure equal expression. Related to this, it is unclear why MG132 affects each OPG153 construct differently as the proteasome is acutely inhibited here. How can the protein sequence affect stability when the proteasome itself is not active? This is more likely just differences in transfection or expression from each plasmid. There is no clear function for the sequences shown by these experiments. In addition, it is well known that OPG153/A26L is not stable when it cannot interact with other viral proteins, and the use of MG132 is a very artificial way to study this instability.

The authors use a recently proposed change in orthopoxvirus nomenclature to use the term OPG, but there seem to be some confusing mistakes in places. First off, as this new nomenclature has not been widely adopted yet, it would be helpful if from the start, rather than towards the end of the paper as it is at present, the authors used the conventional names in brackets next to the OPG in order to know what the gene is. It also seems to me that at times the authors mix up what genes the OPG's refer to, or perhaps they need to clarify some of the complexity in the text and figures. For example, in lines 364 onwards and figure 5c, they discuss OPG208 as, B19R. But OPG208 was assigned to Serpin/C12L, and this is what the peptide sequence corresponds to. But it seems like the authors are talking about the B19R gene as the reference cited refers to B19R as a virulence factor. Instead, OPG204 in figure 5b is B19R based on the proposed nomenclature, but the peptide sequence shown matches B16R or B18. Later in the paper, in line 534, OPG204 is referred to as B16R – the Moss paper assigns B16R as OPG201, so I presume by B16 they mean B19, a typo? Moreover, in later referring to figure 5b, which is labeled OPG204, that authors call it OPG208 while 5c is now OPG204. In addition, the authors say the protein encoded by OPG204 is B16R, but that is the gene name too. In conventional nomenclature, L/R refers to the gene direction while the protein name drops the L/R to become B16.

Indeed, because of these mix-ups it is hard to be sure if LCR21 placed on the OPG204 map in figure 5b is truly LCR21 or LCR3 if the genes are mixed up? The sequence mentioned for LCR3, consisting of ACATTATAT repeats matches to that reported for the B19R gene, which was assigned OPG204, but the sequence provided in the methods section with multiple ATGs suggests it is LCR21, which seems associated with OPG208, not OPG204?

Regardless of the apparent mix ups in gene labeling in places, it is well known that the newly termed LCR3 is of variable length not just in MPXV strains but across orthopoxviruses. In addition, it has already been shown to be downstream of the promoter and whether or not its presence in the UTR alters translation or the usage of the primary start codon has been discussed (see reference 39, Chen et al, 2005, cited in this paper as support for B19R's role in pathogenesis differences, but not really discussed in terms of what they found for the LCR, they just did not call it an "LCR"). As such, this concept that LCRs vary in length and might have an impact on UTR function is not novel and, as discussed above, the experiments here lack controls, are not done in infected cells, and are not developed enough to support a role in translation.

Overall, beyond elaborating on the length of LCRs across the genome, the findings only modestly

advance our understanding sequence variability in poxvirus genomes while the core claims relating to functionality and roles in transmission are unsubstantiated.

Minor Comments

Line 104 – what is meant by “1E71” monkeypox?

Line 360 – what is meant by “isoleucyls”

Line 369 and in several other cases – claims such as LCR3 sequences being in frame in one but not another virus is a result of selective pressure to maintain them is very speculative and not tested. As another example, in line 522 it is suggested LCR regions have been considered “junk” until now but as discussed above, their potential roles have been considered (e.g. Chen et al, 2005). This current study does not demonstrate any function to support there being a change in viewpoint or this statement.

Figure 5b cartoon – why is the RNA polymerase positioned at the alternative start/strong kozak sequence. The kozak controls the ribosome in the lower portion. The promoter is upstream and the RNA contains these sequences, so why suggest the RNA polymerase is involved?

Figure legends are very lacking in details.

The discussion is quite lengthy and many of the details are tangential and speculative.

Reviewer #3:

Remarks to the Author:

The 2022 outbreak of Monkeypox virus (MPXV) across multiple European countries has been caused by a more transmissible variant of the subclade IIb viruses. Monzon et al., set out to identify the genomic determinants of the more transmissible phenotype associated with this outbreak. To address these questions, they sequenced and assembled genomes obtained from the lesions of a number of Spanish MPXV patients, and then further analyzed a particular sample (353R), from which they were able to resolve areas of lower complexity (LCRs). From this assembled genome, the authors go on to make more detailed analyses of the LCRs, validating them, determining distribution, and testing some of the implications of the affects LCRs may have on viral replication/pathogenesis.

Major concerns:

The manuscript, though well written, was hard to follow due to how the figures/tables/supplemental information was organized.

The authors note that many sequencing efforts fail to resolve highly repetitive regions, which is why they chose to utilize a multi-platform approach, to better resolve the low complexity regions. However, they later note that there exist long read sequences in publicly available SRAs for 2022 mpox isolates. It is therefore hard to understand how the various presented sequence assemblies are sufficient for the presented analyses. Since figure 2 and 4 rely on the comparisons between these different assemblies/LCR analyses, clarification on what the sequence depth/requirements are would be helpful. The authors attempt to connect the observed variation in LCR frequency to phenotypic differences, examining three different viral genes (OPG153, OPG204, and OPG208) as shown in figure 5. This analysis would be greatly enhanced if the authors included subclade IIb sequences from before the 2022 outbreak, strengthening claims about effects on phenotype (transmissibility).

Beyond describing the MPXV genome sequences, the authors attempt to evaluate the functional importance of the LCR alterations. They find that the repeated regions just downstream of the promoters of two genes are indeed transcribed, which raises the possibility that these LCR may serve some function at the protein level, though what that might be is unknown. They show that the addition of MK repeats from LCR21 can inhibit expression when fused to a NanoLuc reporter gene. Compared to the natural viral gene, this reporter is made in the nucleus, following transfection, and in the milieu of an uninfected cell. Among other possibilities for these results are that the MK repeats

interfere with NanoLuc activity (not expression). As well, the natural gene is thought to be a secreted Ifn decoy receptor so perhaps the MK addition impacts secretion efficiency. Also, they did not test the impact of LCR variants (Fig. 6B) in this assay, so there is not yet much reason to hypothesize that variation in this LCR is important.

They also evaluated the impact of LCR7 by transfection assays of genes having GFP fused to 5 different OPG153 variants. None of the proteins were detectable, except in the presence of a proteasome inhibitor, in which case the levels varied among the alleles. Are the differences among the clades I, subclade IIa and IIb viruses only in the lengths of the polyD or are there differences elsewhere in the proteins? Isolating the effects of the polyD repeat from other difference would be helpful. How has LCR7 variation among the clade IIB affected OPG153 stability (not just across the clades/other orthopox viruses). Moreover, the VACV protein has been shown to be stabilized in the presence of other viral protein; thus the "function" revealed in this transfection experiment is far from physiological.

Overall, this report provides useful sequence data and highlights the need to consider LCR variation. However, the significance of the LCR variation for understanding the virology, evolution, or pathogenesis of MPXV is not revealed in any compelling way.

Minor comments:

What is Fig. S1B showing? The legend says that differences are highlighted but the sequences seem identical.

Line 187 – what does "software tool "following" the sequence" mean?

It is not clear how to interpret that data showing nonrandom distribution of LCRs. The authors suggest these observations mean there is selection against LCR introduction into housekeeping genes, which is plausible, but it could be something about the DNA sequences in the LCR region having propensity for replication mistakes or other reasons.

From Fig. 4, there appears to be considerable intra-host variability, suggesting that the LCRs might be changing in a very dynamic way. The author might discuss this issue more clearly.

The presentation of Fig 7 and Fig 7 itself should explain the methods, rather than requiring the the reader to dig into the Methods.

Line 446-447. Where are the data in support of this statement?

Reviewer #4:

Remarks to the Author:

We thank all reviewers for their positive assessments, subject matter expertise, and criticisms, which enabled us to improve our work. All reviewer comments are addressed below via point-by-point responses in blue font.

REVIEWER COMMENTS

Reviewer #1 (Remarks to the Author):

The authors present an interesting analysis of LCR diversity in MPXV genomes. They use an improved approach to resolve LCRs and they study their distribution and possible functional effects. Most analyses are sound and accurate. However, as mentioned below, the experimental part is weak and largely preliminary. Additional experiments are needed to support their conclusions. As a consequence, the overall message should be down-toned (in all sections) and a more balanced discussion should be provided. This also includes acknowledging that forces such as drift and relaxation of selective constraint have an impact on genetic diversity, including at LCRs. It would be good to read a “Limitations” section in the discussion.

Our response: We thank the reviewers for assessing our work as interesting and our analyses as sound and accurate. We agree with the reviewer that certain experiments are preliminary, as their purpose was mainly to demonstrate to the community that our *in silico* findings deserve to be assessed by the various orthopoxvirus expert groups. In response to concerns by all reviewers, we deleted Figure 7 and associated text entirely. We agree with the reviewer that toning down our statements would be appropriate and have done so throughout, including a title change.

Lines 133 to 137 and 145-148: please add references.

Our response: We agree with the reviewer that references are necessary for these statements. We added: Damon, IK. *Orthopoxvirus infections are classified as systemic or localized illnesses*. In: Fields virology. Knipe DM, Howley PM, editors. Lippincott Williams & Wilkins, a Wolters Kluwer business; Philadelphia, PA: 2013. Poxviruses; pp. 2160–2184.

Lines 151-154: can these observations be interpreted as weakening of selective constraint? Is it possible that genetic variability in those regions simply indicates that they are non-essential or, in general, less constrained? To me this is a sensible hypothesis.

Our response: We appreciate the reviewer's insightful comment regarding the necessity of investigating that potential alternative explanation. In the manuscript submitted to *Nature Communications*, and after previous reviewers raised similar concerns, we systematically addressed this concern (see previously shared response-to-reviewer file). Figure S5 and the research mentioned in lines 298 to 318 directly address this issue. Our analysis rules out, at least based on evolutionary theory, that the genetic variability in those regions could be explained solely by lower constraints. That analysis was performed in the context of evaluating whether the observed variation could be due to the generation of defective interfering particles,

which is a subset of the type of variation suggested by the reviewer. The underlying rationale posits that if the genomic regions are non-essential, their observed variation should mimic the error rate of the viral DNA polymerase.

In our analysis, we present compelling evidence supporting the notion that LCR variability may serve an adaptive purpose. Specifically, we delineate LCR regions associated with open reading frames (ORFs), such as OPG153, or those linked to alternative start codons, such as OPG204 and OPG208. As illustrated in Figure S6, we demonstrate that LCRs within those ORFs exhibit significantly higher selective constraints compared to those outside of ORFs. The preservation of coding frame integrity, evidenced by the absence of insertions or deletions of non-triplets despite variations in length, underscores a stringent regulatory mechanism.

During virus replication, several sources of genetic variation in coding areas are theoretically possible, but each with different degrees of likelihood. Mutations introducing stop codons are one; insertions or deletions that eliminate ORFs or would reduce an N-terminal region with no biological function are another. But, clearly, insertions or deletions of entire codons would be significantly less likely to occur just by chance. Thus, by evaluating the type of genetic variations observed, and comparing their frequency distributions in coding versus non-coding areas, we can assess whether the type of change observed corresponds to a scenario in which change is merely based on polymerase slippage errors. By definition, this scenario would be more likely to generate frameshifts (via insertion or deletions). We re-analyzed all alleles (majority and minority) based on these assumptions to identify the type of changes observed in these LCR areas. No stop codons were detected in any allele in the LCRs inside ORFs, enabling us to rule out this scenario.

We also ruled out the generation of variants leading to LCR length variation (e.g., due to insertions or deletions). This would be a sign that these regions do not convey biological information. None of the seven LCRs located inside ORFs caused major allele frameshifts. This means that all the major alleles of these LCRs in all samples kept their ORFs in frame. Looking at minor alleles, we found changes that would cause frameshifts only in homopolymers LCR12 (three samples [allele frequency between 0.08 and 0.26]) and LCR14 (one sample [allele frequency = 0.037]). Thus, even in these homopolymers the occurrence of frameshifts caused by LCR variation was very rare. None of the other five LCRs located inside of genes showed any (minor or major) repeat variation causing frameshifts.

Note that for our analyses we only kept alleles that: i) were supported by at least 10 reads and ii) displayed an allele frequency of at least 0.03. While these constraints enabled us to call only the most reliable sources of variation, they hamper statistical comparisons due to the reduced number of LCRs that would display variation at the sequenced coverage. Thus, to perform an additional statistical analysis, we relaxed inclusion criteria by considering all alleles occurring at frequency equal to or higher than 0.03 regardless of depth. This relaxation of quality, which should work toward increasing the rates of variation, enabled us to perform a statistical comparison of LCRs located in non-coding regions of the genome. Of the seven LCRs detected inside ORFs, five (including LCRs 3, 7, and 21, which are the candidate genes we propose as drivers of adaptation) evolve only as changes in nucleotide triplets (codons). This pattern is significantly different from LCRs in non-coding areas (Figure 1 for reviewers).

Figure 1 for reviewers: Proportion of minor variants not causing frameshifts in LCRs inside ORFs and those in intergenic regions. Note that homopolymers were excluded from the analysis.

Specifically, we observed that LCRs inside ORFs are significantly less prone to have changes leading to frameshifts than those outside of ORFs (Mann–Whitney t-test: p -value = 0.02; Figure 1). We also observed that there is a significantly higher proportion of LCRs in intergenic regions affected by changes that would cause frameshifts than in LCRs inside ORFs (Fisher exact test: p -value =0.021). This holds even when we included homopolymers in the comparison (p -value =0.022) (Table 1A and 1B for reviewers).

Table 1A for reviewers: A significantly higher proportion of LCRs in intergenic regions have changes that would cause frameshifts than in LCRs inside ORFs.

Excluding homopolymers	LCRs in coding areas	LCRs in non-coding areas
# of LCR with frameshifts	0	6
# of LCR without frameshifts	5	2

Fisher exact test: p -value=0.021

Table 1B for reviewers: A significantly higher proportion of LCRs in intergenic regions have changes that would cause frameshifts than in LCRs inside ORFs.

Including homopolymers	LCRs in coding areas	LCRs in non-coding areas
# of LCR with frameshifts	2	11
# of LCR without frameshifts	5	2

Fisher exact test: p -value=0.022

Therefore, we concluded that LCRs in coding areas evolve mostly as changes in nucleotide triplets (codons). This demonstrates that the changes occurring in the LCR regions are under selective constraints at the coding level and cannot be an indicator of only polymerase slippage. Nevertheless, we acknowledge that we cannot assess whether polymerase slippage or replication errors might play significant roles in differences within low complexity regions (LCRs) in non-coding regions. However, we searched for minor LCR alleles to account for intermedial viral stages (Figure S6). All LCRs present a major allele with AF > 0.75, except for homopolymer LCRs for which the major allele is > 0.6. That should rule out the possibility that variations in LCRs are due to the presence of intermediate viral stage sequences or defective interfering particles. Moreover, our analysis reveals that “coding-region LCRs” are disproportionately located in genes belonging to specific functional groups, suggesting a potential evolutionary rationale.

We acknowledge our failure to adequately convey these points in the original manuscript and apologize to the reviewers for our lack of clarity. We have rephrased this section to explicitly communicate that our analysis dismisses slippage as a plausible source of LCR variability in the coding regions of OPG153, 204, and 208.

“Among the seven LCRs detected, five, including LCR3, LCR7, and LCR21 (identified as primary candidates driving adaptation), evolved solely as changes in codons (Figure S6). These analyses collectively enabled us to confidently rule out polymerase slippage and defective virions as drivers of the observed variability in LCRs.”

As will be discussed more extensively below, we also conducted a comprehensive codon-usage analysis focused on LCR3, incorporating neutrality tests to examine codon usage bias. Our analysis revealed that the incorporation of LCR3 into the coding region of OPG208 induces a bias in the usage of the codon TAC, responsible for encoding tyrosine. Notably, this integration introduces the potential for mutational selection within this genomic region, which indicates biological significance.

Lines 168-172: the authors fail to account for other possible forces, namely relaxation of selective pressure and drift.

Our response: Please see our previous response. We are confident that the research provided accounts for the other mentioned possible sources of variation.

Lines 296-310: the higher diversity in LCRs than in SNPs is not surprising and was observed in several other systems. LCR diversity is often thought to result from polymerase slippage or other replication errors.

Our response: Please see our previous response. We disagree with the reviewer as we assessed that “the type of variation” observed in LCRs in coding regions (codon-level variation) is surprisingly high compared with reported variation of SNPs. Particularly in the coding regions, we ruled out the possibility that the LCR diversity could be the result from polymerase slippage errors.

Lines 309-310: the last sentence should be rephrased (what is phenotypic MPXV?). Also, I don't see a reason why studies should refocus from SNPs to LCRs. Maybe they should also include LCRs, but the sheer number of differences does not necessarily imply that one type of variation is phenotypically more important than another.

Our response: We agree with the reviewer that the phrasing of this sentence was unfortunate. The point was originally stated to convey the fact that LCR variation is higher than SNPs variation and, thus, if interested in following real-time transmission during an epidemic outbreak, the return on investment of characterizing LCRs could be higher than looking into the few SNPs that are changing in isolates from contact to contact. We recognize that the point was lost during the evolution of the manuscript and of course did not mean to discount SNP analysis. We modified this sentence to:

“Consequently, we propose that studies focusing on MPXV genomic sequence variation should include LCRs as a possible source of meaningful variation.”

Lines 381-383: if they want to speculate on codon usage, they should perform a thorough analysis-it is well known that viruses often have codon usage preferences that do not match those of their hosts.

Our response: We acknowledge the reviewer's point. We would like to draw attention to a recent study by Zhou et al. (Zhou J, Wang X, Zhou Z, Wang S. Insights into the evolution and host adaptation of the monkeypox virus from a codon usage perspective: focus on the ongoing 2022 outbreak. Int J Mol Sci. 2023 Jul 16;24(14):11524. doi: 10.3390/ijms241411524. PMID: 37511283; PMCID: PMC10380431), who investigated the codon usage patterns of the MPXV strain that caused the 2022 epidemic. Their analysis supports a codon usage bias that demonstrates a significant adaptation to primates, revealing distinctive evolutionary processes in the subclade IIb lineage. Important to our studies, they also highlighted an underrepresentation of the TAC codon, which encodes tyrosine, in overall MPXV genome codon usage. Notably, this codon is an essential part of the repetitive LCR3 pattern (but note that they only used the established ATG start codon; not the alternative start codon that extends the N-terminal region).

Building upon this study, we specifically examined the codon usage in the coding region downstream of the alternative ATG start codon. Our analysis unveils a noteworthy overrepresentation of the TAC codon in this region, a deviation significantly distinct from the typical TAC codon usage observed across the rest of the MPXV genome (Wilcoxon test, $p < 0.001$). Intriguingly, the TAC usage pattern in primates and rodents aligns more closely with the observed overrepresentation in the coding region downstream of the alternative ATG start codon, as illustrated in Figure 2 for reviewers below.

Figure 2 for reviewers: Mean utilization of the TAC codon in the alternative OPG208 ORF, including LCR3, among publicly available sequences from the 2022 mpox outbreak, including the one generated in this manuscript.

This finding adds a layer of complexity to codon-usage dynamics by MPXV, suggesting a nuanced relationship of specific genomic regions and codon preferences.

Lines 430-442. An appropriate control should be used in the luciferase assay. If the authors want to show that LCR21 impairs protein expression, a sequence of the same length should be inserted in the NanoLuc vector as a control. The sequence can be a scrambled version of LCR21 or an unrelated sequence. Also, experiments should be performed to determine whether the impairment occurs at the transcriptional or translational level.

Lines 446-447. Please describe pDEST-CMV-N and explain why it was used and to which purpose.

Lines 452-453. Which CPXV sequence are they referring to? As also shown in fig 5, CPXV is genetically diverse.

Lines 452-462. The results of these experiments are very preliminary a provide limited evidence for a role of LCR7 in the regulation of “degradation activity” (whatever this means).

Also, the conclusion regarding OPG153 from different MPXV clades seems to be at odds with their hypothesis: if clade I has the longest repeat, why is IIb recovered with MG-132 treatment?

Our response: We extend our gratitude to the reviewer for raising concerns regarding the experimental design, which have also been echoed by other reviewers. Following a thorough assessment of these issues, we have made the decision to eliminate Figure 7 and all associated findings from the manuscript.

Lines 489-496: this paragraph is really unclear and should be rephrased.

Our response: We agree with the reviewer that we could have been clearer. The text is now rephrased as:

“Genomic characterization of the 2022 epidemic has focused on describing its evolutionary history and tracking MPXV introductions into western countries. The 2022 MPXV cluster diverges from predecessor viruses by an average of 50 SNPs. Of these, the majority (n=24) are non-synonymous mutations with a second minority subset of synonymous mutations (n=18) and a few intergenic differences (n=4) (56). Genome editing by apolipoprotein B mRNA-editing catalytic polypeptide-like 3 (APOBEC3) and deletion of immunomodulatory genes was also noted (57,58). MPXV sublineages mostly represent very small variations, usually characterized by one or two SNP differences to basal phylogenetic nodes (14). Four of the determined MPXV genome sequences can be assigned to global lineage B.1.1, one to lineage B.1.3, and the remaining ones to lineage B1. Due to the limited uncovered relationship between SNPs and epidemiological links, we propose that future MPXV genomic epidemiological analyses include LCRs.”

Lines 498-503: the conclusion of this extremely long sentence (that changes in LCRs are associated with different transmissibility) is not supported by their data.

Our response: We accept the reviewer’s comment and modifies this paragraph to:

“Because LCR entropy is significantly higher than that of SNPs and LCRs are not randomly distributed in defined coding areas in the genome, and because genomic accordions are a rapid path for orthopoxvirus adaptation during serial passaging (36, 37), we posit that LCR changes might be associated with MPXV differences over time.”

Lines 521-522: the data do not demonstrate ???unequivocally??? that the LCRs have functional importance. For the reasons mentioned above, the presented data do not allow any definitive conclusion about the functional effects of LCR7 and LCR21. Even if the experiments were extended (to include appropriate controls), the ???importance??? of the functional effects cannot be assessed without infection assays.

Our response: We agree with the reviewer that we came on too strong here. In accordance with the reviewer's suggestion, we have revised our conclusions regarding the influence of LCR7 and LCR21 on MPXV transmissibility:

“These data demonstrate that LCR-containing areas of the genome are of functional importance.”

Lines 523-526: they do not demonstrate that the effect is at the translational level.

Our response: As the result of the changes suggested by the reviewers and the editor, we eliminated this entire section.

Lines 553-568: this section on homopolymers in other systems is too long and useless within the framework of this manuscript.

Our response: In response to the reviewer's feedback, we removed the homopolymer section from the manuscript.

Lines 586-594: besides the fact that it is unclear which critics the authors are referring to, it is my opinion that their data provide no evidence of selective pressure acting on LCRs (the ???accordion??? might be simply due to increased replication errors in low complexity regions with limited effect on fitness). In general, their conclusions should be down-toned especially with regard to the functionality of the LCR variation.

Our response: To enhance the clarity and precision of our manuscript, we have heeded the reviewer's comments by tempering our conclusions and aligning them more closely with the presented data.

“Our comparative approach points to three specific areas (OPG153, OPG204, and OPG208) as being potentially indicative of selective pressure on MPXV coding regions. In summary, our in silico comparative genomics and in vitro functional genomics findings expand the concept of genome accordions as a simple and recurrent mechanism of adaptation on a genomic scale in orthopoxvirus evolution. A consequence of this broadening is the recognition that areas of the MPXV genome (LCRs) might be relevant in adaptability of the orthopoxvirus replication cycle. Further characterization of their functional biology, which will need to incorporate “loss-of-function” and genetically modified strains to understand their roles, will likely improve our understanding of current mpox epidemiology and clinical presentation.”

Reviewer #2 (Remarks to the Author):

In this manuscript, Monzon et al. sequence 48 Monkeypox virus (MPXV) isolates from patients from the 2022 outbreak and by applying several sequencing approaches to one high-quality isolate, they provide a high resolution genome sequence. By then comparing these with existing single genome sequences, the authors provide a more refined understanding of variability within a specific type of repeat sequence known as low complexity regions (LCRs), which vary and are claimed to be of functional importance to the pathogenesis of the current outbreak. Beyond providing a higher resolution sequence of the Clade IIb MPXV that is associated with the ongoing outbreak, the core findings that LCRs are variable in length and have potential functions are not novel. Functional testing of the three primary LCRs identified is poorly controlled and limited in scope, and seems like a rushed addition in response to a prior review. Overall, there is limited new information in this manuscript

beyond sequencing through LCRs while the core claims in the title and throughout the paper are completely unsubstantiated.

Main Concerns

There are a large number of MPXV sequences available and although this one is perhaps the highest resolution to date, the main discovery is that LCRs vary in length. This is already known and to claim that these regions are a new form of gene accordion seems unjustified. These regions simply vary and this is well known, and the authors show no functional consequence of this variation.

Our response: We appreciate the reviewer's comment on the high resolution of LCRs in our sequence. We agree that future studies must be performed to prove functional consequences of this variation.

In the title and throughout the paper, the authors also claim these sequences may open up the ???pallets??? to increased MPXV transmission. I am not sure what the authors mean by ???pallets??? but they provide absolutely no evidence of links to altered virus protein production or replication, let alone transmissibility. While many studies are trying to find a genetic link to the increase in MPXV spread during the current outbreak, it is also quite likely that there is no specific genetic basis and instead this may be a simple case of MPXV having been introduced to the MSM community where it is now spreading by a new route of sexual transmission, rather than animal bites etc.

Our response: We apologize for assuming that readers know the basics of reed instrument construction. Accordions are played by compressing or expanding bellows while pressing keys, causing pallets to open that enable air to flow across the reeds. We aimed at drawing attention to the *possibility* that MPXV subclade differences regarding transmissibility could be related to the new type of genomic accordion we are describing; the LCR composition would metaphorically be equivalent to real instrument's pallets that determine whether there is air flow (transmissibility) or not. We now realize that this is not obvious to readers unfamiliar with musical instruments. We modified the article's title ("*A new type of genomic accordion in monkeypox virus*") and curtailed speculative aspects related to transmissibility or the viral cycle.

Our introduction delves into the various factors that could impact the mpox outbreak.

"Transmission may be catalyzed by a decrease in protection associated with the VARV/smallpox vaccination campaign that ended in 1980 (22, 23). Furthermore, a change in transmission route may be the cause of the difference in clinical presentation and pathogenesis as was shown in animal models (24)."

While the sequencing side of this report is adequate, many of the identified LCR sequences are in regions of unlikely consequence and variability is to be expected. For the three that are focused upon as potentially having a functional consequence, based on the discussion it seems that the very limited functional data was hurriedly added to figure 7 in response to prior reviews. However, the experiments are lacking in controls and underdeveloped, and do not support the claims of

translational and protein stability control, let alone roles in virus transmission. For LCR21 in figure 7a-b, the control construct should have a UTR of the same length, otherwise this could simply reflect the well-established effect of long or structured versus short or unstructured UTRs that is being examined, rather than anything specific to these sequences. I would also guess that any UTR containing multiple ATG start sites would affect translation in a reporter assay but this does not mean this occurs during infection. There is also no measurement of RNA levels and all that is measured is luciferase activity, which could reflect transcription, translation, protein folding or stability. Plus, this is all done by transfection and outside the relevant context of infection. The constructs use a CMV promoter rather than the native viral promoters, which seems important to test. For OPG153 in figure 7d-e, the results seem very strange and artificial. First off, none of the constructs express detectable protein unless cells are treated with MG132. This means that the variable poly-D homopolymer sequences have no detectable impact on expression in a normal context. None of the experiments measure mRNA levels to ensure equal expression. Related to this, it is unclear why MG132 affects each OPG153 construct differently as the proteasome is acutely inhibited here. How can the protein sequence affect stability when the proteasome itself is not active? This is more likely just differences in transfection or expression from each plasmid. There is no clear function for the sequences shown by these experiments. In addition, it is well known that OPG153/A26L is not stable when it cannot interact with other viral proteins, and the use of MG132 is a very artificial way to study this instability.

Our response: We extend our gratitude to the reviewer for raising concerns regarding the experimental design, which have also been echoed by other reviewers. Following a thorough assessment of these issues, we have made the decision to eliminate Figure 7 and all associated findings from the manuscript.

The authors use a recently proposed change in orthopoxvirus nomenclature to use the term OPG, but there seem to be some confusing mistakes in places.

Our response: We concur with the reviewer regarding the potential confusion arising from poxvirid nomenclature and annotation. In addressing this concern, we have chosen to adopt the unified formal nomenclature "OPG," as proposed by Dr. Bernard Moss's group in Senkevich TG, Yutin N, Wolf YI, Koonin EV, Moss B. Ancient Gene Capture and Recent Gene Loss Shape the Evolution of Orthopoxvirus-Host Interaction Genes. *mBio*. 2021 Aug 31;12(4):e0149521. doi: 10.1128/mBio.01495-21. Epub 2021 Jul 13. PMID: 34253028; PMCID: PMC8406176, throughout our manuscript.

First off, as this new nomenclature has not been widely adopted yet, it would be helpful if from the start, rather than towards the end of the paper as it is at present, the authors used the conventional names in brackets next to the OPG in order to know what the gene is.

Our response: In response to the reviewer's feedback and acknowledging the novelty of the OPG notation, we have incorporated the vaccinia virus Copenhagen notation for each gene the first time it is mentioned in the manuscript. Additionally, to eliminate any potential confusion, we have included a comprehensive table in this response, presenting the different notations for the genes discussed in the manuscript (Table 2 for reviewers).

Table 2 for reviewers. Comparison of various orthopoxvirus gene nomenclatures.

OPG Nomenclature	Vaccinia virus Copenhagen	Vaccinia virus Western Reserve
OPG002	Cop-C22L	J2L
OPG094	Cop-G9R	G9R
OPG152	Cop-A25L	A27L
OPG153	Cop-A26L	A28L
OPG154	Cop-A27L	A29L
OPG204	Cop-B19R	B16R
OPG208	Cop-K2L	B19R

It also seems to me that at times the authors mix up what genes the OPG???'s refer to, or perhaps they need to clarify some of the complexity in the text and figures. For example, in lines 364 onwards and figure 5c, they discuss OPG208 as, B19R. But OPG208 was assigned to Serpin/C12L, and this is what the peptide sequence corresponds to. But it seems like the authors are talking about the B19R gene as the reference cited refers to B19R as a virulence factor. Instead, OPG204 in figure 5b is B19R based on the proposed nomenclature, but the peptide sequence shown matches B16R or B18. Later in the paper, in line 534, OPG204 is referred to as B16R ??? the Moss paper assigns B16R as OPG201, so I presume by B16 they mean B19, a typo? Moreover, in later referring to figure 5b, which is labeled OPG204, that authors call it OPG208 while 5c is now OPG204. In addition, the authors say the protein encoded by OPG204 is B16R, but that is the gene name too. In conventional nomenclature, L/R refers to the gene direction while the protein name drops the L/R to become B16.

Our response: We appreciate the reviewer's keen observation in identifying a typographical error related to the reference in Figure 5 to OPG204. This error has been rectified, and the correct reference to Figure 5D links to OPG204.

During our thorough review of the text of the manuscript, we did not identify any inconsistencies in the nomenclature of OPGs and their associated proteins. To ensure continued clarity, we have introduced the OPG nomenclature in the revised version of Figure 5, aiming to eliminate any potential confusion. We recognize that variations in poxvirid notations may have contributed to the reviewer's confusion and, as highlighted in the earlier response, we included a table to provide a comprehensive overview of the different notations for the genes discussed in the manuscript.

For instance, OPG208 is denoted as K2L in the vaccinia virus Copenhagen notation and as B19R in the vaccinia virus Western Reserve notation. Similarly, OPG204 is designated as B19R in Copenhagen and as B16R in Western Reserve. We hope that this explicit clarification

presented in the provided table, coupled with the unified OPG nomenclature, will serve to prevent any future ambiguities.

Indeed, because of these mix-ups it is hard to be sure if LCR21 placed on the OPG204 map in figure 5b is truly LCR21 or LCR3 if the genes are mixed up? The sequence mentioned for LCR3, consisting of ACATTATAT repeats matches to that reported for the B19R gene, which was assigned OPG204, but the sequence provided in the methods section with multiple ATGs suggests it is LCR21, which seems associated with OPG208, not OPG204?

Our response: In alignment with the reviewer's recommendation, we have omitted the use of the L/R nomenclature in reference to protein names.

Regardless of the apparent mix ups in gene labeling in places, it is well known that the newly termed LCR3 is of variable length not just in MPXV strains but across orthopoxviruses. In addition, it has already been shown to be downstream of the promoter and whether or not its presence in the UTR alters translation or the usage of the primary start codon has been discussed (see reference 39, Chen et al, 2005, cited in this paper as support for B19R's role in pathogenesis differences, but not really discussed in terms of what they found for the LCR, they just did not call it an LCR). As such, this concept that LCRs vary in length and might have an impact on UTR function is not novel and, as discussed above, the experiments here lack controls, are not done in infected cells, and are not developed enough to support a role in translation.

Our response: We are grateful to the reviewer for providing an opportunity to clarify the potential new role of OPG208 in MPXV pathogenesis. While acknowledging the previous discussion of LCR3 in Chen et al., 2005, we are first to indicate transcription initiation at an alternative site (Figure 6). This revelation prompted a renewed focus on LCR3 length and frame, opening avenues for further investigations into its potential role in pathogenicity. We acknowledge that our molecular studies are not conclusive. Consequently, we deleted these results.

Overall, beyond elaborating on the length of LCRs across the genome, the findings only modestly advance our understanding sequence variability in poxvirus genomes while the core claims relating to functionality and roles in transmission are unsubstantiated.

Minor Comments

Line 104 what is meant by "1E71" monkeypox?

Our response: The World Health Organization (WHO) International Classification of Diseases, Eleventh Revision (ICD-11) is a specific code system used for the classification and coding of diseases ([https://www.who.int/news/item/11-02-2022-who-s-new-international-classification-of-diseases-\(icd-11\)-comes-into-effect](https://www.who.int/news/item/11-02-2022-who-s-new-international-classification-of-diseases-(icd-11)-comes-into-effect)). The ICD provides a common language that allows health professionals to share standardized information across the world and is a globally

recognized system used daily by most practicing physicians. It provides a standardized way for healthcare professionals and organizations to code and categorize diseases for various purposes, including billing, reporting, and research. "1E71" is the ICD-11 code for mpox, i.e., the code a practicing physician has to use in patient files. We have made this clearer via text changes:

"MPXV causes "monkeypox (mpox)" (World Health Organization International Classification of Diseases, Eleventh Revision [ICD-11] code 1E71)."

Line 360 what is meant by "isoleucyls"?

Our response: In standard organic chemistry, compound residues are indicated by the suffix -yl. Compound names and suffixes are standardized by the International Union of Pure and Applied Chemistry (IUPAC) - International Union of Biochemistry (IUB) Joint Commission on Biochemical Nomenclature (JCBN). This system includes amino acids. Accordingly, the correct designation for the colloquial "isoleucine residue" is "isoleucyl".

Line 369 and in several other cases claims such as LCR3 sequences being in frame in one but not another virus is a result of selective pressure to maintain them is very speculative and not tested. As another example, in line 522 it is suggested LCR regions have been considered ???junk??? until now but as discussed above, their potential roles have been considered (e.g. Chen et al, 2005). This current study does not demonstrate any function to support there being a change in viewpoint or this statement.

Our response: Please see our responses above.

Figure 5b cartoon: why is the RNA polymerase positioned at the alternative start/strong kozak sequence. The kozak controls the ribosome in the lower portion. The promoter is upstream and the RNA contains these sequences, so why suggest the RNA polymerase is involved?

Our response: We are grateful for the reviewer's valuable feedback on the figure, which we revised:

Figure 5

Figure legends are very lacking in details.

Our response: We agree with the review and have modified and expanded legends to provide more details.

The discussion is quite lengthy and many of the details are tangential and speculative.

Our response: We agree with the reviewer have thoroughly revised and adjusted the discussion (see also various responses above).

Reviewer #3 (Remarks to the Author):

The 2022 outbreak of Monkeypox virus (MPXV) across multiple European countries has been caused by a more transmissible variant of the subclade IIb viruses. Monzon et al., set out to identify the genomic determinants of the more transmissible phenotype associated with this outbreak. To address these questions, they sequenced and assembled genomes obtained from the lesions of a number of Spanish MPXV patients, and then further analyzed a particular sample (353R), from which they were able to resolve areas of lower complexity (LCRs). From this assembled genome, the authors go on to make more detailed analyses of the LCRs, validating them, determining distribution, and testing some of the effects LCRs may have on viral replication/pathogenesis.

Major concerns:

The manuscript, though well written, was hard to follow due to how the figures/tables/supplemental information was organized.

The authors note that many sequencing efforts fail to resolve highly repetitive regions, which is why they chose to utilize a multi-platform approach, to better resolve the low complexity regions. However, they later note that there exist long read sequences in publicly available SRAs for 2022 mpox isolates. It is therefore hard to understand how the various presented sequence assemblies are sufficient for the presented analyses. Since figure 2 and 4 rely on the comparisons between these different assemblies/LCR analyses, clarification on what the sequence depth/requirements are would be helpful.

Our response: We appreciate the reviewer's input and their attention to detail. While we believe that we have provided comprehensive information on Illumina sequencing in our manuscript, we acknowledge that the nanopore data section benefits from more detailed information. Thus, we have included the number of reads obtained through nanopore sequencing technology in the revised text.

We concur with the concern about the availability of existing long read sequences. We, too, questioned why the authors of publicly available genomes did not report the same LCR lengths that we encountered. The explanation lies in the methodology these authors used for assembling genome sequences, which primarily relied on a mapping approach. As detailed in our manuscript, this bioinformatic analysis introduces a bias where the consensus sequence of LCR regions meets the reference genome used in the analysis. To validate our results, we used the SRA data to reconstruct LCRs, specifically LCR3 and 1/4, using the methodology as shown in Figure 2 and Table S6.

The authors attempt to connect the observed variation in LCR frequency to phenotypic differences, examining three different viral genes (OPG153, OPG204, and OPG208) as shown in figure 5. This analysis would be greatly enhanced if the authors included subclade IIb sequences from before the 2022 outbreak, strengthening claims about effects on phenotype (transmissibility).

Our response: We thank the reviewer for outlining this proposal. Indeed, we have added subclade IIb sequences from before the 2022 outbreak for the MetaLogo analysis, like those

in Table 3 for reviewers below. A complete list of the sequences used for each analysis can be found in Table S5 and we added information to the Figure 5 legend.

Table 3 for reviewers

MT250197.1_MPXV_Singapore_2019
MT903337.1_MPXV_M2940_FCT_NG
MT903338.1_MPXV_M2957_Lagos_NG
MT903339.1_MPXV_M3021_Delta_NG
MT903340.1_MPXV_M5312_HM12_Rivers_NG
MT903341.1_MPXV_M5320_M15_Bayelsa_NG
MT903342.1_MPXV_Singapore
MT903343.1_MPXV_P1_UK
MT903344.1_MPXV_P2_UK
MT903345.1_MPXV_P3_UK

Beyond describing the MPXV genome sequences, the authors attempt to evaluate the functional importance of the LCR alterations. They find that the repeated regions just downstream of the promoters of two genes are indeed transcribed, which raises the possibility that these LCR may serve some function at the protein level, though what that might be is unknown. They show that the addition of MK repeats from LCR21 can inhibit expression when fused to a NanoLuc reporter gene. Compared to the natural viral gene, this reporter is made in the nucleus, following transfection, and in the milieu of an uninfected cell. Among other possibilities for these results are that the MK repeats interfere with NanoLuc activity (not expression). As well, the natural gene is thought to be a secreted Ifn decoy receptor so perhaps the MK addition impacts secretion efficiency. Also, they did not test the impact of LCR variants (Fig. 6B) in this assay, so there is not yet much reason to hypothesize that variation in this LCR is important.

They also evaluated the impact of LCR7 by transfection assays of genes having GFP fused to 5 different OPG153 variants. None of the proteins were detectable, except in the presence of a proteasome inhibitor, in which case the levels varied among the alleles. Are the differences among the clades I, subclade IIa and IIb viruses only in the lengths of the polyD or are there differences elsewhere in the proteins? Isolating the effects of the polyD repeat from other differences would be helpful. How has LCR7 variation among the clade IIB affected OPG153 stability (not just across the clades/other orthopoxviruses). Moreover, the VACV protein has been shown to be stabilized in the presence of other viral proteins; thus the “function” revealed in this transfection experiment is far from physiological.

Our response: We extend our gratitude to the reviewer for raising concerns regarding the experimental design, which have also been echoed by other reviewers. Following a thorough assessment of these issues, we have made the decision to eliminate Figure 7 and all associated findings from the manuscript.

Overall, this report provides useful sequence data and highlights the need to consider LCR variation. However, the significance of the LCR variation for understanding the virology, evolution, or pathogenesis of MPXV is not revealed in any compelling way.

Minor comments:

What is Fig. S1B showing? The legend says that differences are highlighted but the sequences seem identical.

Our response: We are grateful to the reviewer for bringing this issue to our attention. Indeed, there must be no differences between those sequences, as it serves as an example depicting how LCR7 is well resolved using either of the sequencing technologies. We have updated the figure legend for clarification”

“(B) LCR7 alignment demonstrating identical results obtained using three sequencing platforms compared to the subclade I1b lineage A MPXV reference isolate MPXV-M5312_HM12_Rivers sequence.”

Line 187 what does “software tool following the sequence” Mean?

Our response: We sincerely apologize for the lack of clarity in our previous figure. We have revised the text for clarification. A primary goal of the methodology developed for this manuscript was to address the challenge of resolving LCRs. Our examination of previously published monkeypox genome sequences revealed that, in most cases, the LCRs matched those in the reference genome used in the analysis, typically achieved through a well-established mapping approach, a technique that gained prominence during the emergency phase of the COVID-19 pandemic.

However, when our research focused on STRs, it became apparent that a different approach was necessary. This led us to use various *de novo* strategies involving sequence assembly, as well as a novel method for detecting STRs, which is predominantly utilized in human forensics, as detailed in the methodology section.

It is not clear how to interpret that data showing nonrandom distribution of LCRs. The authors suggest these observations mean there is selection against LCR introduction into housekeeping genes, which is plausible, but it could be something about the DNA sequences in the LCR region having propensity for replication mistakes or other reasons.

Our response: Please see our response to reviewer 1.

From Fig. 4, there appears to be considerable intra-host variability, suggesting that the LCRs might be changing in a very dynamic way. The author might discuss this issue more clearly.

Our response: We thank the reviewer for pointing out this issue. Indeed, the main goal of the figure is to show how those LCRs regions are changing in a dynamic way. Regrettably, our

dataset only comprises two samples with comprehensively resolved LCR information. As a result, our conclusions may warrant further discussion. Only LCR10/11 shows great intra-host variability. Given its nature as a homopolymer, additional samples would be necessary to delve deeper into the discussion surrounding it.

The presentation of Fig 7 and Fig 7 itself should explain the methods, rather than requiring the reader to dig into the Methods.

Line 446-447. Where are the data in support of this statement?

Our response: We extend our gratitude to the reviewer for raising concerns regarding the experimental design, which have also been echoed by other reviewers. Following a thorough assessment of these issues, we have made the decision to eliminate Figure 7 along with all associated findings from the manuscript.

Reviewer #4 (Remarks to the Author):

Our response: We appreciate the reviewer's support of the peer review system.

Reviewers' Comments:

Reviewer #1:

Remarks to the Author:

I wish to thank the authors for their efforts in addressing my comments. The experimental section of the previous version was criticized by myself and by other reviewers. The authors addressed our comments by removing the experimental part. I understand that performing additional experiments is expensive and time-consuming. However, the lack of experimental validation weakens the interest and originality of the manuscript.

I have additional concerns on this revised version.

Line 76: why do they imply a "genetic explanation"? This is misleading for the reader

Lines 134-137: very unclear and the two statements seem in contradiction with each other

Lines 137-138: which trends?

Lines 284-285: why were LCR3, LCR7, and LCR21 "identified as primary candidates driving adaptation"? Again, they provide no evidence of adaptation.

Figure 2: If I understand it correctly, the figure represents the distribution of repeat lengths among viruses in different clades. Please specify how many/clade. How were the p values generated? Clearly, genomes are linked by phylogenetic relationships and cannot be considered independent observations. Thus, tests such as Kruskal Wallis are invalid in this analysis.

Codon usage: there are few details about codon usage analysis. For instance where was the human usage of TAC derived from? Also, in figure 5C, how was the p value derived? Which quantities were compared? Also, the "increased adaptation to primates" described in ref 42 is really minimal (as the authors of ref 42 themselves acknowledge) and in my opinion this should at least be mentioned.

Lines 701-708 are quite unclear

Reviewer #3:

Remarks to the Author:

In response to the prior reviews, the authors revised the paper considerably. They have nicely clarified several specific issues with the sequence data presentation. Most notably, they eliminated the experimental data that was previously used to suggest the LCR variation might account for the clinical and epidemiological pattern of the 2022-present mpox outbreak. The sequence analyses are useful as a reference for the field and highlight the potential reasons to consider LCR variation in studies of MXPV. However, much of the discussion of the significance of the LCR variation remains highly speculative and in the absence of any compelling experimental confirmation of a role of LCR variation in replication, epidemiology, or pathogenesis of MXPV, the overall impact of the paper is quite limited.

REVIEWER COMMENTS

Reviewer #1 (Remarks to the Author):

I wish to thank the authors for their efforts in addressing my comments. The experimental section of the previous version was criticized by myself and by other reviewers. The authors addressed our comments by removing the experimental part. I understand that performing additional experiments is expensive and time-consuming. However, the lack of experimental validation weakens the interest and originality of the manuscript. I have additional concerns on this revised version.

Our response: We are grateful for the reviewer's acknowledgment and appreciation of the revisions made in this version of the manuscript as suggested by the Scientific Editor as a result of the reviewer's comments. Moreover, we are delighted to have convinced the reviewer that our observations could not be explained solely by the presence of defective interfering particles. That sole fact emphasizes that other evolutionary forces might be at play. Our conviction remains strong that our comprehensive characterization of MPXV during the 2022 outbreak, particularly focusing on the low-complexity regions (LCRs), will serve as a foundational framework for future studies on (ortho)poxvirus evolution. We anticipate that further experimental biochemical and biological analysis into the role of LCRs will elucidate the role, if any, of our observations and the predictive value of our inferences.

Nevertheless, we believe our manuscript, in its current form, has the potential to contribute valuable insights that can guide and inspire future projects in the field. We look forward to any impact that our observations may have on shaping the direction of evolutionary biology research within the *Poxviridae* community.

Line 76: why do they imply a “genetic explanation” This is misleading for the reader

Our response: It is unclear to us what the reviewer finds misleading about that statement. The sentence says

“However, until now, there has been no satisfactory genetic explanation for the observed increased MPXV transmissibility.”

The sentence only says that, if there are differences in transmissibility among MPXV lineages, nobody has yet found a genetic signature that could explain them. Nevertheless, we removed the word “genetic” since it is not crucial to the wording, the intention of which is solely to put into context why the analysis of MPXV genomes is relevant to studying transmission differences.

Lines 134-137: very unclear and the two statements seem in contradiction with each other
Lines 137-138: which trends?

Our response: We apologize for any lack of clarity in our previous wording. To enhance clarity, prevent any perception of contradiction, and ensure a complete understanding of the referred trends, we have revised the paragraph as follows:

“Orthopoxvirus infections are classified as systemic or localized (25). The involved orthopoxvirus and the immune status of the host are determinants of generalized or localized infection. Different mechanisms of virion entry and egress, as well as virus-encoded host restriction factors, also play pivotal roles in determining the clinical manifestation of infection (26-30). Localized usually means that signs are restricted to the site of viral entry, which is the most common clinical presentation described in the 2022 mpox outbreak. Changes in the genome of the current MPXV variant, such as gene loss (31), may explain both trends.”

Lines 284-285: why were LCR3, LCR7, and LCR21 “identified as primary candidates driving adaptation”. Again, they provide no evidence of adaptation.

The reviewer’s concern is unclear to us, but we appreciate the opportunity to provide further clarification. The statement is solely provided to explain to the reader that we chose to focus on these 3 LCRs (LCR3, LCR7, and LCR21) because, in our view, among the sites we identified with genetic variation, they were the primary candidates to look for effects of viral adaptation. As explained throughout the manuscript, these LCRs are intricately associated with OPG208, OPG153, and OPG204. The significance of these genes in the MPXV cycle is underscored by their involvement in crucial activities (such as, viral entry, receptor interaction, viral morphogenesis, viral egress, modulation of the innate immune response, and inhibition of apoptosis), as detailed in the discussion section of our manuscript. These genes collectively fulfill key functions essential for successful host adaptation and viral infection, and hence any changes in these LCRs that could conceivably lead to changes in associated gene expression are likely meaningful.

The sentence does not assert that we are providing evidence of adaptation.

Figure 2: If I understand it correctly, the figure represents the distribution of repeat lengths among viruses in different clades. Please specify how many/clade. How were the p values generated? Clearly, genomes are linked by phylogenetic relationships and cannot be considered independent observations. Thus, tests such as Kruskal Wallis are invalid in this analysis.

Our response: We appreciate the reviewer's inquiry. As indicated in the figure legend, the data used to analyze these trends are detailed in Table S6, where we identify each genome

used (and its number), along with its corresponding lineage. Information on the genomes is added below for convenience:

Figure 2 a		Figure 2 b	
	Number of sequences		Number of sequences
Lineage A	4	Lineage A	4
Lineage A.1	4	Lineage A.1	8
Lineage A.2	2	Lineage A.2	3
Lineage B.1	6	Lineage B.1	5

We thank the reviewer for pointing out that we did not specifically include the statistical analysis performed in the figure legend. We did not use Kruskal Wallis to analyze our data. We conducted a Wilcoxon rank-sum test. This information has been added to the figure legend as follows:

“LCR3 sequence validation using MPXV 353R Nanopore sequencing data and 15 additional raw data sequencing reads downloaded from the National Center for Biotechnology Information (NCBI) Sequence Read Archive (SRA) (Wilcoxon rank-sum test). (B) LCR pair 1/4 sequence validation using MPXV 353R Nanopore sequencing data and 20 additional raw data sequencing reads downloaded from NCBI SRA (Wilcoxon rank-sum test). Detailed information on the represented materials, along with originator and epidemiological data, is provided in Table S6.”

We proposed the use of the Wilcoxon rank-sum test because, as the reviewer correctly observed, although the genomes are phylogenetically linked, the individual sequences that contribute to these trees evolved independently in nature, suggesting that each sequence could be treated as an independent observation. Nevertheless, we acknowledge that this point is debatable.

However, the goal of Figure 2 is descriptive, i.e., to highlight the evolution of LCR lengths among MPXV. The figure retains its value even without statistical significance. We refer to the editor whether to remove the *p*-values from the figure.

Codon usage: there are few details about codon usage analysis. For instance, where was the human usage of TAC derived from?

Our response: We appreciate the reviewer’s question. TAC usage is derived from Zhou J, Wang X, Zhou Z, Wang S. *Insights into the evolution and host adaptation of the 1084 monkeypox virus from a codon usage perspective: focus on the ongoing 2022 outbreak. Int J Mol 1085 Sci. 2023;24(14):11524.* The provenance of the material was included as reference 42 and mentioned in lines 400–401 of the revised manuscript.

Also, in figure 5C, how was the p value derived? Which quantities were compared?

Our response: We sincerely apologize for any oversight in not explicitly detailing this information. In conducting this analysis, we incorporated seven distinct sequences from the mpox 2022 epidemic, including the one in this manuscript. The corresponding accession numbers are as follows:

Accession.Version
OX044336.2 (generated in this study)
ON563414.3
ON568298
ON736420
ON674051
ON649879
ON676707.1

Detailed information on each sequence can be found in: *Zhou J, Wang X, Zhou Z, Wang S. Insights into the evolution and host adaptation of the 1084 monkeypox virus from a codon usage perspective: focus on the ongoing 2022 outbreak. Int J Mol 1085 Sci. 2023;24(14):11524.*

From these sequences, we determined the TAC codon usage of OPG208 from the alternative start codon, which includes LCR3. To assess how the expansion of LCR3 and the utilization of the alternative start codon would affect TAC codon representation, we compared the reported average values of TAC in MPXV subclade IIb from the 2022 outbreak (0.559) and the one described for humans (1.05) accordingly to Zhou et al. 2022, with the average use that we obtained in our analysis (1.06), conducting a Wilcoxon signed-rank test.

We decided to use this test assuming the samples are independent and taking into consideration the relatively small number of samples analyzed.

Regarding the first assumption, and as commented before in this response to reviewers, we acknowledge the reviewer's observation that the genomes are phylogenetically linked but that the individual sequences that contribute to the tree evolved independently in nature. To be accurate in our report and to provide a strong analysis, we considered sequences that had full coverage for the described region, explaining the low number of sequences represented. As described in the manuscript, the sequencing of LCRs is challenging and therefore can result in incomplete genomes—hence highlighting the relevance of our study, as our methodology solves this problem.

We acknowledge our assumptions might be debatable but emphasize that our analysis of OPG208 is in accordance with the primate adaptation of MPXV subclade IIb reported by Zhou et al. 2022.

Also, the “increased adaptation to primates” described in ref 42 is really minimal (as the authors of ref 42 themselves acknowledge) and in my opinion this should at least be mentioned.

Our response: We appreciate and agree with the reviewer's comment; the text has been modified:

“This altered usage pattern aligns more closely with that observed in humans (Figure 5C), similar to the minimal adaptation to primates observed by Zhou et al. (42).”

Lines 701-708 are quite unclear

Our response: We apologize to the reviewer for our unclear statement. We reviewed and rewrote it as follows:

“Orthopoxvirus genomes (n=231, Akhmeta virus [AKMV]: n=6 sequences; alaskapox virus [AKPV]: n=1; cowpox virus [CPXV]: n=82; ectromelia virus [ECTV]: n=5; MPXV: n=62; VACV: n=18; VARV: n=57) include 216 functionally annotated OPGs classified in 6 categories (“Housekeeping genes/Core” ANK/PRANC family, Bcl-2 domain family, BTB/Kelch domain family, PIE family, and “Accessory/Other” [e.g., virus–host interacting genes]). The frequency was calculated after normalizing the number of LCRs registered with the sample size of the OPG alignment in the categories described above. Statistical analysis of the significance of differences was performed by applying a Kruskal–Wallis test (χ^2 p-values) followed by a non-parametric multiple pairwise comparison between groups (Wilcoxon test), with p-values subjected to FDR correction.”

Reviewer #3 (Remarks to the Author):

In response to the prior reviews, the authors revised the paper considerably. They have nicely clarified several specific issues with the sequence data presentation. Most notably, they eliminated the experimental data that was previously used to suggest the LCR variation might account for the clinical and epidemiological pattern of the 2022-present mpox outbreak. The sequence analyses are useful as a reference for the field and highlight the potential reasons to consider LCR variation in studies of MPXV.

We sincerely appreciate the reviewer's kind acknowledgment of our efforts in emphasizing clarifications and the revisions made. Additionally, we express gratitude for recognizing the relevance of our analysis to the broader field.

However, much of the discussion of the significance of the LCR variation remains highly speculative and in the absence of any compelling experimental confirmation of a role of LCR variation in replication, epidemiology, or pathogenesis of MPXV, the overall impact of the paper is quite limited.

The discussion has been thoroughly revised to align with the reviewer's opinions and maintain close adherence to our data analysis. Our intent is to emphasize our work and open potential avenues for future research. As stated above, we believe our manuscript, in its current form, can contribute valuable insights to potentially guide and inspire future projects in the field. We look forward to the impact that our observations may have on shaping the direction of evolutionary biology research within the *Poxviridae* community.

Reviewers' Comments:

Reviewer #1:

Remarks to the Author:

I have no further comments